# Cellular senescence triggers intracellular acidification and lysosomal pH alkalinized via ATP6AP2 attenuation in breast cancer cells

Wei Li [1], Kosuke Kawaguchi [1✉], Sunao Tanaka[1], Chenfeng He[1], Yurina Maeshima[1], Eiji Suzuki[2] & Masakazu Toi [1]

Several chemotherapeutic drugs induce senescence in cancer cells; however, the mechanisms underlying intracellular pH dysregulation in senescent cells remain unclear. Adenosine tri-phosphatase H$^+$ transporting accessory protein 2 (ATP6AP2) plays a critical role in maintaining pH homeostasis in cellular compartments. Herein, we report the regulatory role of ATP6AP2 in senescent breast cancer cells treated with doxorubicin (Doxo) and abemaciclib (Abe). A decline in ATP6AP2 triggers aberrant pH levels that impair lysosomal function and cause immune profile changes in senescent breast cancer cells. Doxo and Abe elicited a stable senescent phenotype and altered the expression of senescence-related genes. Additionally, senescent cells show altered inflammatory and immune transcriptional profiles due to reprogramming of the senescence-associated secretory phenotype. These findings elucidate ATP6AP2-mediated cellular pH regulation and suggest a potential link in immune profile alteration during therapy-induced senescence in breast cancer cells, providing insights into the mechanisms involved in the senescence response to anticancer therapy.

[1] Department of Breast Surgery, Kyoto University Graduate School of Medicine, 54 Shogoin-kawaharacho, Sakyo-ku, Kyoto 606-8507, Japan. [2] Kobe City Medical Center General Hospital, 2-1-1 Minatojimaminami-cho, Chuo-ku, Kobe 650-0047, Japan. ✉email: kkosuke@kuhp.kyoto-u.ac.jp

Breast cancer is a highly prevalent and substantial cause of mortality in women worldwide[1]. Neoadjuvant chemotherapy and adjuvant systemic therapy improve the prognosis and increase the survival of breast cancer patients[2]. Despite advances in breast cancer treatment, many patients still experience resistance to therapy and exhibit an unfavorable prognosis[3].

Cellular senescence is a physiological stress response in normal and cancerous cells and is characterized by irreversible cell cycle arrest and permanent proliferation suppression[4,5]. Several factors, including excessive activation of oncogenes, accumulation of stress, metabolic disturbances, irradiation, and chemotherapeutic drugs, can lead to cellular senescence[5–8]. However, the specific role of senescence in the development of cancer remains unclear. Extensive research has shown that senescence suppresses neoplastic formation and tumor progression by restricting the repair and proliferation mechanisms of impaired cells[9–11]. Similarly, senescence inhibits tumor progression by enhancing the antitumor immune response[12,13]. Senescence-induced therapies trigger senescence-associated secretory phenotype (SASP)-dependent vascular remodeling to sensitize cells to cytotoxic therapeutic drugs[14]. Senescence enhances antitumor immune responses by promoting the clearance of senescent cells by immune cells. However, recent evidence has established that senescent cells secrete multiple pro-inflammatory factors through the SASP, which contribute to immune evasion, creating a double-edged sword effect[15–17]. Therefore, it is unclear whether therapy-induced senescence reprograms the SASP to affect inflammatory and immune responses.

Adenosine triphosphatase H$^+$ transporting accessory protein 2 (ATP6AP2), also known as the prorenin receptor, is an essential accessory subunit for the biogenesis of active vacuolar-type adenosine triphosphatase (V-ATPase)[18–21]. V-ATPases are complex multi-subunit enzymes that function as rotary nanomotors that pump protons and maintain intracellular pH (pH$_i$) homeostasis[22–24]. V-ATPase defects perturb autophagy in various systems[25]. ATP6AP2 is predominantly localized in the lysosome and plasma membrane, and plays a crucial role in energy conservation and acidification of intracellular compartments to support cellular biological activity[26,27]. Moreover, several studies have shown that ATP6AP2 interacts with transforming growth factor (TGF)-β1 and Wnt/β-catenin molecules, which regulate the biology process during embryonic development, stem cell differentiation, and tumor formation[28–30]. ATP6AP2 promotes the tumorigenesis and progression of many cancer types[31–33]. ATP6AP2 is a vital protein involved in fundamental cellular processes, and its ablation results in impaired viability due to multiple organ deficiencies[34]. Conditionally ablated ATP6AP2 has further elucidated its critical role in preserving cellular homeostasis across various cell types[35–37]. However, the function of ATP6AP2 in regulating pH$_i$ homeostasis in doxorubicin (Doxo) and abemaciclib (Abe)-challenged breast cancer is unclear.

Recent studies have suggested that an impaired pH$_i$ is a hallmark of cancer[38]. Dysregulated pH$_i$ dynamics facilitate various cancer cell behaviors such as cell proliferation, migration, metastasis, evasion of apoptosis, and metabolic adaptation[39–41]. Additionally, an acidified intracellular environment suppresses antibody-dependent cytotoxicity in breast cancer cells[42]. Therefore, linking cellular senescence and intracellular acidification, and further exploring their molecular mechanisms, may provide novel insights into breast cancer treatment strategies. Despite numerous metabolic reprogramming events occurring in senescent cells, which result in modifications to the intracellular environment, it is unknown whether senescence triggers aberrant changes in pH$_i$ and lysosomal pH (pH$_L$) in breast cancer. Elucidating the mechanisms underlying these changes is essential for understanding the implications of senescence in cancer biology and guiding potential therapeutic interventions.

Here, we showed that intracellular acidification occurs in senescent breast cancer cells along with abnormal pH$_L$. Therefore, we hypothesized that a specific molecule drives pH$_i$ and pH$_L$ aberrations in therapy-induced senescent breast cancer cells. To explore this possibility, we first induced senescence in breast cancer cells using two cycles of therapy in vitro with Doxo and Abe. To further elucidate the effect of therapy-induced senescence in breast cancer cells, we used bulk RNA sequencing (RNA-seq) to reveal transcriptome-wide changes in senescent and non-senescent cells. Among the identified differentially expressed genes (DEGs) with overlapping downregulated expression after senescence, functional analysis highlighted the role of V-ATPase. Therefore, we explored the effect of V-ATPase and its specific subunits on the therapy-induced senescence of breast cancer cells. Among these subunits, we demonstrated that *ATP6AP2* was downregulated in therapy-induced senescent breast cancer cells, causing intracellular acidification (reduction in pH$_i$) and lysosomal alkalinization (elevation in pH$_L$). Additionally, attenuation of *ATP6AP2* expression during therapy-induced senescence triggers SASP reprogramming to activate various pro-inflammatory factors that regulate inflammation and immune profile alterations.

## Results

**Doxo and Abe promote cellular senescence accompanied by an altered profile of senescence-related genes in breast cancer cells.** Doxo and Abe were used to treat breast cancer cells (human triple-negative breast cancer cell line MDA-MB-231 and human luminal A subtype breast cancer cell line MCF-7) for 24 h, without a robust cytotoxic effect. After the cells were grown in Doxo- and Abe-free medium for 48 h, they were re-treated with these two therapeutic reagents. Various features of cellular senescence were observed after 24, 48, 72, and 96 h (Fig. 1a). The Cell Cycle Kit-8 (CCK-8) assay showed that Doxo significantly suppressed the proliferation of both MDA-MB-231 and MCF-7 cells (Supplementary Fig. 1a). Abe inhibited cell proliferation in time- and dose-dependent manners (Supplementary Fig. 1b). However, 1000 nM Abe caused massive cell death, and 24 h of treatment was insufficient to suppress proliferation; thus, this concentration of Abe was deemed unsuitable for senescence induction in breast cancer cells. To investigate whether proliferation was suppressed through cell cycle arrest under our treatment conditions, we performed flow cytometry to analyze cell cycle distribution in breast cancer cells. Doxo induced G2 cell cycle arrest in MDA-MB-231 cells, in contrast to the G1 phase arrest observed in MCF-7 cells (Supplementary Fig. 1c, d). Conversely, Abe induced G1 cell cycle arrest in both MDA-MB-231 and MCF-7 cells at various concentrations (Supplementary Fig. 2a–f). These results suggested that Doxo and Abe substantially inhibited breast cancer cell proliferation, an essential feature of cellular senescence, through cell cycle arrest in a time- and dose-dependent manner.

We next determined whether Doxo and Abe triggered cellular senescence in breast cancer cells based on senescence-associated β-galactosidase (SA-β-Gal) staining, a widely used senescence marker. Doxo triggered significant cellular senescence after 48 h of exposure and showed more pronounced staining with further exposure, particularly after 96 h of treatment (Fig. 1b, d). Similarly, treatment with 500 nM Abe increased the proportion of senescent cells and caused significant cellular senescence after 96 h of exposure (Fig. 1c, e). Both 250 and 125 nM Abe induced cellular senescence, but the effect was more pronounced at 250 nM, where the proportion of senescent cells was higher than that in the 125 nM treatment group, which showed a relatively

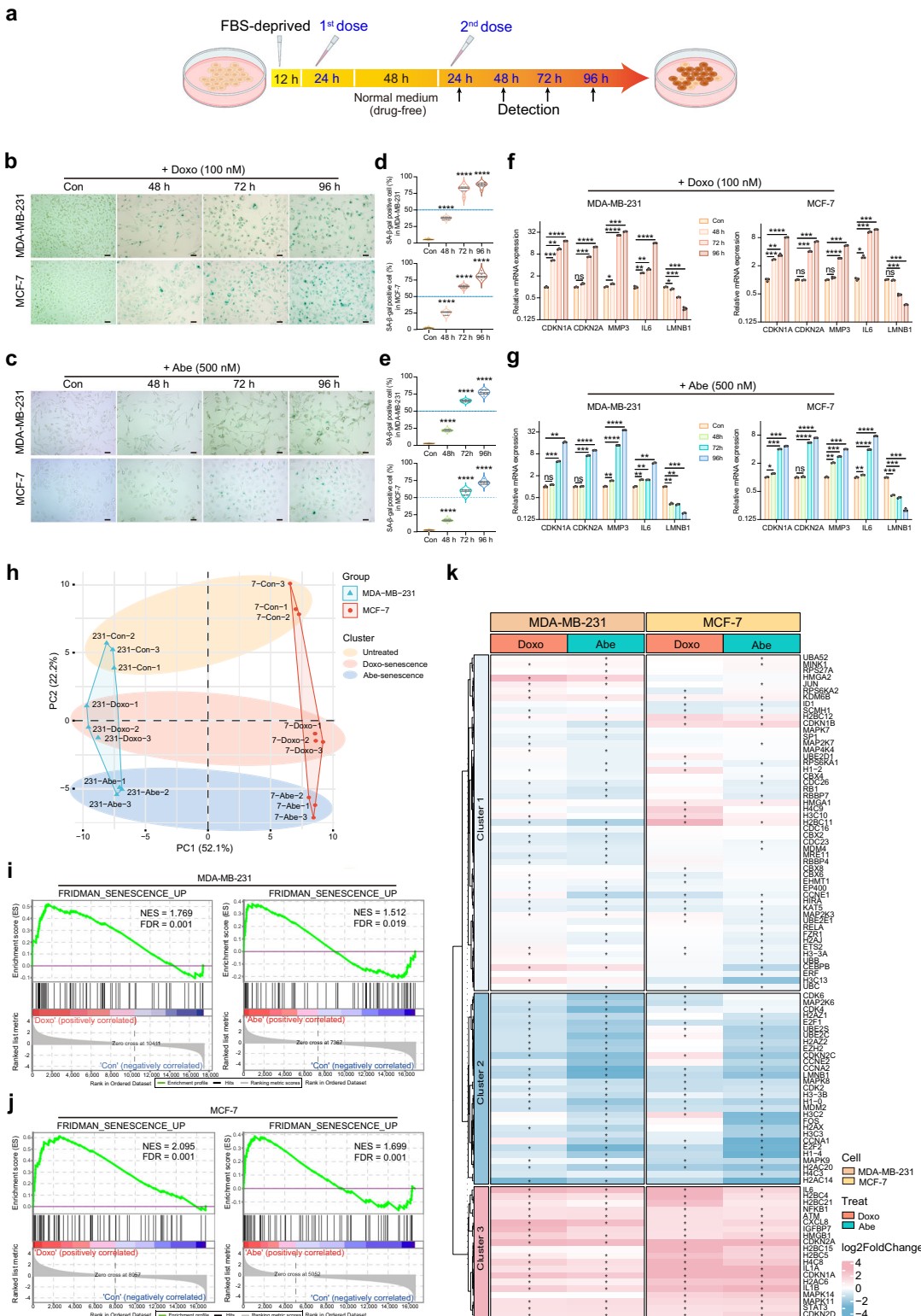

lower proportion of senescent cells (less than 50%). (Supplementary Fig. 3a–d). Moreover, the microscopic images demonstrated that the cell morphology became irregular, and that cell size was enlarged in the senescent cells compared to that in the control non-senescent cells.

Considering the essential roles of numerous molecular expression profiles correlated with the senescence phenotype, we next sought to verify the molecules considered hallmarks of cellular senescence. Reverse transcription-quantitative

polymerase chain reaction (RT-qPCR) revealed that Doxo simultaneously increased the mRNA levels of cyclin-dependent kinase inhibitor 1A (CDKN1A), cyclin-dependent kinase inhibitor 2A (CDKN2A), matrix metalloproteinase-3 (MMP3), and interleukin-6 (IL6), accompanied by decreased laminin subunit beta-1 (LMNB1) levels, with significant changes detected after 96 h of treatment (Fig. 1f). Similarly, 500 nM Abe induced significant changes in the expression of senescence-related molecules, with the most prominent effect observed after 96 h

**Fig. 1 Doxo and Abe promote cellular senescence accompanied by an altered profile of senescence-related genes in breast cancer cells. a** Experimental design and analysis workflow of cellular senescence induced by Doxo and Abe in breast cancer cells (created with BioRender.com). **b, c** Representative SA-β-Gal staining images in therapy-challenged MDA-MB-231 and MCF-7 cells treated with doxorubicin (Doxo; 100 nM) (**b**) and abemaciclib (Abe; 500 nM) (**c**). **d, e** Quantitative analysis of the percentage of SA-β-Gal–positive cells in therapy-challenged breast cancer cells treated with Doxo (100 nM) (**d**) and Abe (500 nM) (**e**). At least six separate fields of view were analyzed. **f, g** RT-qPCR detection of relative mRNA expression levels of senescence-related genes (*CDKN1A, CDKN2A, MMP3, IL6, LMNB1*) in Doxo (100 nM)-treated (**f**) and Abe (500 nM)-treated (**g**) breast cancer cells. **h** Principal components analysis (PCA) based on senescence-related gene expression profile using "REACTOME_CELLULAR_SENESCENCE" gene set. **i, j** Gene set enrichment analysis (GSEA) was performed in Doxo- and Abe-treated MDA-MB-231 (**i**) and MCF-7 (**j**) cells compared with control groups, respectively. Genes (vertical black lines) represented in gene sets are on the *x*-axis, and the *y*-axis represents the enrichment score (ES). The green line connects genes and ES points. The colored band shows the degree of correlation of genes with the enriched phenotype (red; positive correlation, and blue; negative correlation). False discovery rate (FDR) < 0.05 as the significance threshold. **k** Heatmap represents cellular senescence-related gene profiles in Doxo- and Abe-treated breast cancer cells. asterisk indicates a statistically significant difference in gene expression. Scale bars represent 50 μm. Data are shown as the means ± SD of three independent experiments. Statistical analyses were performed with one-way ANOVA with Dunnett's multiple comparisons test (**d**, **e**) and two-way ANOVA with Dunnett's multiple comparisons test (**f**, **g**). ns, not significant; *$P < 0.05$, **$P < 0.01$, ***$P < 0.001$, ****$P < 0.0001$.

of exposure (Fig. 1g). Moreover, 250 nM Abe treatment yielded molecular changes with a similar trend to 500 nM Abe treatment (Supplementary Fig. 3f), whereas alterations in the expression levels of senescence-associated molecules induced by 125 nM Abe treatment were unstable (Supplementary Fig. 3e). Thus, 125 nM Abe treatment was excluded from subsequent experiments.

We further analyzed and profiled the gene expression of therapy-induced cellular senescence using bioinformatic methods applied to bulk RNA-seq. Principal component analysis (PCA) revealed an obvious difference in a distinct cluster of differentially expressed senescence-related genes between therapy-challenged and untreated breast cancer cells (Fig. 1h). Additionally, Gene Set Enrichment Analysis (GSEA) indicated that the Doxo and Abe treatment groups were enriched in the "FRIDMAN_SENES-CENCE-UP" gene set (Fig. 1i, j). The heatmap illustrates the expression patterns of senescence-related genes (using the "REACTOME_CELLULAR_SENESCENCE" gene set) obtained from bulk RNA-seq analysis (Fig. 1k). Overall, these results demonstrated that Doxo and Abe trigger stable cellular senescence by simultaneously upregulating *CDKN1A* and *CDKN2A* expression. Furthermore, the senescent phenotype became increasingly evident with prolonged treatment, particularly after 96 h.

**Doxo and Abe induce V-ATPase functional suppression in senescent breast cancer cells.** To explore the phenotypic changes induced by Doxo and Abe treatment in senescent breast cancer cells, we compared the transcriptional alterations between senescent and control cells. Among the DEGs, 29 genes were found to be commonly downregulated in both cell lines, as shown in the Venn diagram in Fig. 2a and Supplementary Data 1–4. Subsequently, we performed Gene Ontology (GO) enrichment analysis to determine the biological functions of the down-regulated genes. GO enrichment analysis revealed that these genes were predominantly enriched in the GO biological processes (BP) "cellular senescence" and "intracellular pH reduction," in the cellular compartment (CC) "vacuolar proton-transporting V-type ATPase complex," and in the molecular function (MF) "proton−transporting ATPase activity, rotational mechanism" (Fig. 2b). Therefore, we hypothesized that therapy-induced cellular senescence causes a decrease in V-ATPase, which consequently contributes to $pH_i$ decline.

To ascertain whether V-ATPase attenuates the therapy-triggered $pH_i$ decrease in breast cancer cells, we used 200 nM bafilomycin A1 (Baf A1), a specific V-ATPase inhibitor, to treat MDA-MB-231 and MCF-7 cells over a time gradient. After cells were subjected to 1 h of 200 nM Baf A1 treatment, the $pH_i$ was observed to be lower than that of control cells, and a significant difference in $pH_i$ was detected after 3 h of inhibitor treatment

(Supplementary Fig. 4a). Notably, 200 nM Baf A1 treatment did not affect cell viability (Supplementary Fig. 4b). Fluorescence microscopy confirmed that the $pH_i$ decreased under the same treatment conditions (Fig. 2c, d).

As shown in Fig. 2b, the overlapping downregulated genes were also enriched in the GO-BP term associated with lysosomal membrane. To further elucidate whether V-ATPase attenuated the Doxo- and Abe-induced increase in $pH_L$, we assayed $pH_L$ in breast cancer cells treated with 200 nM Baf A1 for 3 h. Baf A1 significantly decreased yellow fluorescence (emission [Em] 535 nm) intensity, with a corresponding increase in blue fluorescence (Em 440 nm) intensity in both MDA-MB-231 (Fig. 2e, f) and MCF-7 (Fig. 2g, h) cells. This shift in the fluorescence spectra resulted in a substantial decrease in the ratio of yellow to blue fluorescence intensity, indicating an increase in $pH_L$. These consistent changes in the fluorescence profiles were observed in both MDA-MB-231 and MCF-7 cells following Baf A1 treatment. Therefore, Doxo and Abe impede the function of V-ATPase, thereby inducing the dysregulation of cellular pH homeostasis in senescent breast cancer cells.

**Doxo and Abe trigger substantial intracellular acidification in senescent cells.** To further investigate whether Doxo- and Abe-induced senescence in breast cancer cells affects $pH_i$, we identified a significant negative correlation between Doxo and Abe treatment groups and cellular pH regulation using GSEA (Fig. 3a, b). We then performed $pH_i$ detection using pHrodo Green AM staining, a pH-sensitive fluorescent probe, at various time points. Doxo elicited intracellular acidification after 72 h of exposure and this effect was most pronounced at 96 h (Fig. 3c–f). Similarly, 500 nM Abe generated significant intracellular acidification in MDA-MB-231 cells at 72 h, particularly at 96 h (Fig. 3g, upper panel, and Fig. 3h). Moreover, treatment of MCF-7 cells with 500 nM Abe not only resulted in marked intracellular acidification at 72 and 96 h exposure, but also induced a certain extent of $pH_i$ decrease at 48 h of exposure (Fig. 3g, lower panel, and Fig. 3i). Similarly, treatment with 250 nM Abe triggered intracellular acidification after 72 and 96 h of exposure (Supplementary Fig. 5a–d).

Furthermore, we assessed the proportion of senescent cells and the corresponding $pH_i$ values under the same treatment conditions to better understand this correlation. Strong positive correlations were observed between senescence and $pH_i$ in MDA-MB-231 ($r^2 = 0.9533$) and MCF-7 ($r^2 = 0.9649$) cells (Fig. 3j). In addition, cells with changes in size and morphology showed relatively higher fluorescence intensity. These results imply that Doxo and Abe induced significant intracellular acidification, and a more pronounced decrease in $pH_i$ substantially affected the induction of cellular senescence.

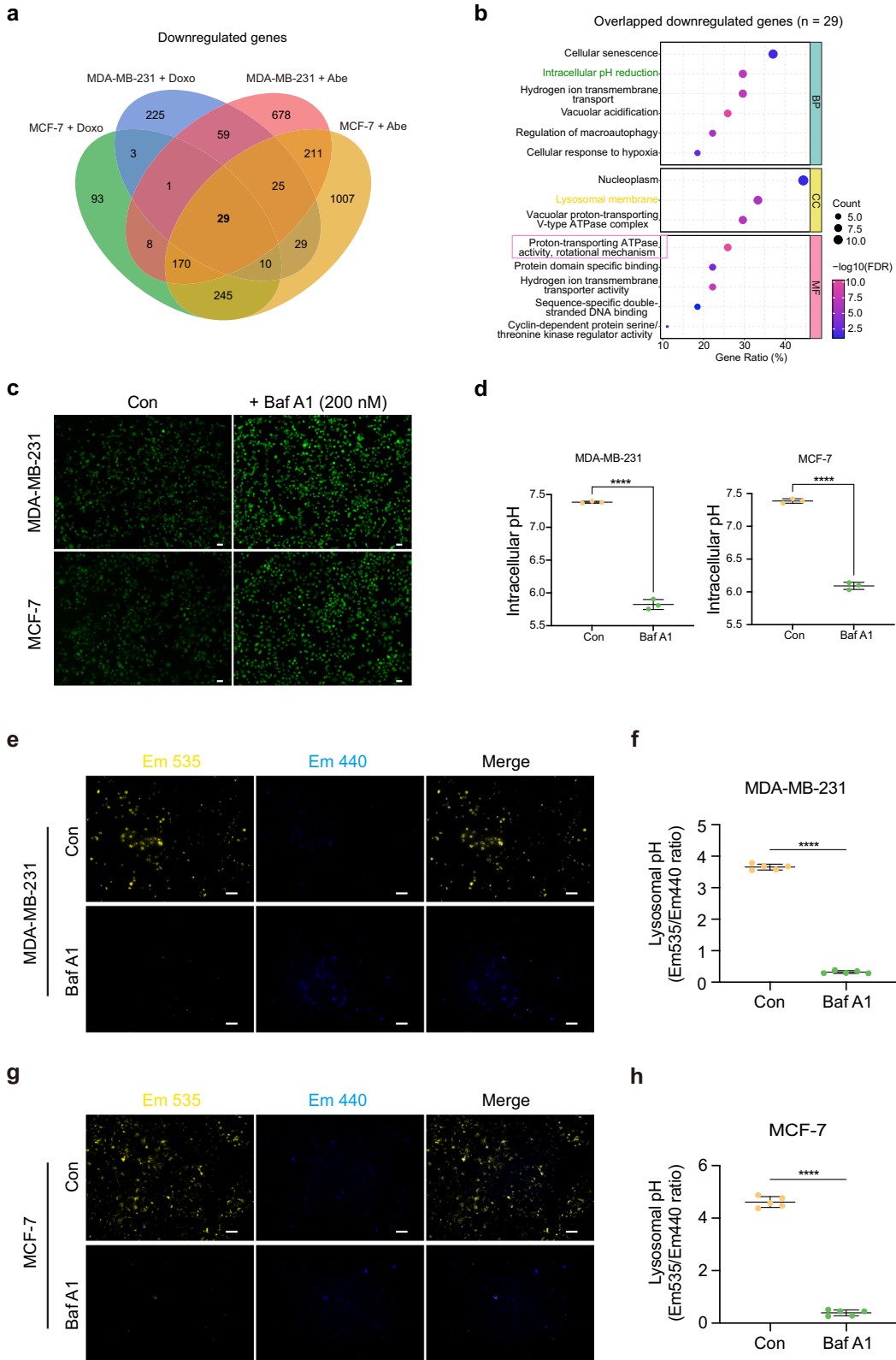

**Doxo and Abe elicit an alkalinized pH_L to cause lysosomal impairment in senescent breast cancer cells**. RNA-seq data showed that DEGs in therapy-induced senescent cells were enriched in the lysosomal membrane (Fig. 2b). To further elucidate the potential connection between pH_L changes and cellular senescence, we employed a Sankey plot to illustrate the functional enrichment analysis of the overlapping downregulated genes

($n = 29$) clustered in the senescence and lysosome terms (Fig. 4a). Furthermore, the differentially expressed V-ATPase subunits were enriched in lysosomal terms.

Subsequently, we assessed pH_L status using the LysoSensor Yellow/Blue DND-160 probe. For these experiments, the cells were subjected to 72 and 96 h of exposure, as the most distinct cellular senescence and notable intracellular acidification were

**Fig. 2 Doxo and Abe induce V-ATPase functional suppression in senescent breast cancer cells. a** Four-ellipse Venn diagram outlining overlapped downregulated genes in various therapy-challenged breast cancer cells. **b** Gene Ontology (GO) enrichment analysis of overlapped downregulated genes ($n = 29$) shown with biological process (BP), cellular component (CC), and molecular function (MF) terms. All represented GO terms showed significant enrichment at an FDR < 0.05. The color represents the statistical significance of the term. The circle size indicates the counts of enriched genes. **c, d** Representative fluorescent images of breast cancer cells treated with Bafilomycin A1 (Baf A1; 200 nM) for 3 h stained with pHrodo Green AM (**c**) and quantitative analysis of the intracellular pH (pH$_i$) (**d**). **e, f** Representative fluorescent images and quantitative analysis of lysosomal pH (pH$_L$) in Baf A1 (200 nM)-treated MDA-MB-231 cells stained with LysoSensor Yellow/Blue-DND-160. **g, h** Representative fluorescent images and quantitative analysis of pH$_L$ in Baf A1 (200 nM)-treated MCF-7 cells. Emission (Em) 535 nm (yellow); Em 440 nm (blue). Scale bars represent 50 μm. Data are shown as the means ± SD of three independent experiments. Data were analyzed by unpaired two-tailed Student's $t$ test (**d, f, h**) and Fisher's exact test with Benjamini–Hochberg multiple-testing correction (**b**). ****$P < 0.0001$.

observed under these conditions. Doxo treatment markedly decreased yellow fluorescence intensity (emission at 535 nm), along with a corresponding increase in blue fluorescence intensity (emission at 440 nm) in MDA-MB-231 cells, resulting in a marked reduction in the fluorescence intensity ratio from yellow to blue. (Fig. 4b, c). Similarly, Doxo induced pH$_L$ elevation in MCF-7 cells after 72 and 96 h of exposure (Fig. 4d, e). These results suggest that Doxo increased pH$_L$ in breast cancer cells after 72 and 96 h of treatment. pH$_L$ exhibited a comparable increase (based on a decrease in the yellow/blue ratio) in breast cancer cells treated with 500 nM Abe (Fig. 4g, i). However, in contrast to the effects of Doxo treatment, 500 nM Abe treatment appeared to enhance blue fluorescence rather than weaken yellow fluorescence in MDA-MB-231 and MCF-7 cells (Fig. 4f, h). Additionally, pH$_L$ increased significantly in MCF-7 cells treated with 250 nM Abe, but only significantly increased after 96 h of treatment in MDA-MB-231 cells (Supplementary Fig. 6a, b).

Given that lysosomes are acidic membrane-bound organelles that depend on the acidic pH environment in the lysosome lumen to maintain normal function in biological processes[43,44], we proceeded to determine whether therapy-induced aberrant pH$_L$ potentially leads to lysosomal dysfunction. The mRNA expression of lysosomal-associated membrane protein 2 (*LAMP2*), the main membrane protein of lysosomes[45,46], was significantly suppressed in breast cancer cells after Doxo treatment at both 72 and 96 h (Fig. 4j). Similarly, 500 nM Abe down-regulated *LAMP2* expression in breast cancer cells (Fig. 4k). In contrast, 250 nM Abe significantly decreased *LAMP2* expression at 96 h of treatment in MDA-MB-231 cells but at both 72 and 96 h of exposure in MCF-7 cells (Supplementary Fig. 6c). Furthermore, bulk RNA-seq data confirmed that Doxo and Abe downregulated lysosome-related genes compared to control cells (Supplementary Fig. 6d and Supplementary Table 1). Notably, these results revealed that the decrease in *LAMP2* expression was strongly correlated with pH$_L$ status under the same treatment conditions.

To gain a deeper insight into the relationship between therapy-induced cellular senescence and pH$_L$ status, we evaluated the proportion of senescent cells and their corresponding pH$_L$ levels in MDA-MB-231 ($r^2 = 0.9660$) and MCF-7 ($r^2 = 0.9843$) cells under identical treatment conditions (Fig. 4l). A robust correlation was observed between cellular senescence and pH$_L$ levels, suggesting a functional link between these two phenotypes. Thus, we concluded that therapy-induced senescence caused an alkaline shift in pH$_L$, and this abnormal pH$_L$ environment, coupled with *LAMP2* suppression, triggered lysosomal impairment.

**Doxo and Abe attenuate ATP6AP2 expression in senescent breast cancer cells.** We further confirmed that the hub gene was responsible for compromised V-ATPase activity and disrupted cellular pH homeostasis. The volcano plot illustrates the DEGs identified in the bulk of the gene expression matrix. Notably, *ATP6AP2* was consistently significantly downregulated in the Doxo- and Abe-treated groups (Fig. 5a). Intriguingly, in both

MDA-MB-231 and MCF-7 cells, the majority of DEGs were upregulated following treatment with Doxo, but downregulated after treatment with Abe, compared to the expression levels in control cells. We first analyzed the expression levels of V-ATPase subunits among DEGs in non-senescent cells and found a relatively higher baseline expression of *ATP6V0E1*, *ATP6V1F*, *ATP6AP2*, and *ATP6V1G1* (Supplementary Fig. 7a, b). Subsequently, we visualized the expression profile of the differentially expressed V-ATPase subunits using a heatmap (Fig. 5b and Supplementary Table 2).

We identified a significant upregulation in *ATP6AP2* expression among the tumor groups by analyzing The Cancer Genome Atlas Breast Invasive Carcinoma (TCGA-BRCA) data (Fig. 5c). Moreover, we observed a positive correlation between *ATP6AP2* and CD8$^+$ T cell tumor immune infiltration through the analysis of the Tumor Immune Estimation Resource (TIMER) database (Fig. 5d). Analysis of the top four subunits of V-ATPase based on their basal expression levels revealed that *ATP6V0E1* and *ATP6V1F* exhibited relatively weak positive correlations with CD8$^+$ T cell infiltration in tumors. Conversely, *ATP6V1F* negatively correlated with CD8$^+$ T cell infiltration (Supplementary Fig. 7c).

To further elucidate the association between *ATP6AP2* and lysosomal function, we performed a protein–protein interaction (PPI) analysis involving differentially expressed V-ATPase subunits and lysosome-related genes. The analysis revealed significant interactions among various differentially expressed V-ATPase subunits. Notably, *ATP6AP2* displayed a pronounced association with *LAMP2*, indicating the potential functional relevance between *ATP6AP2* and lysosomal processes (Fig. 5e).

Following the validation of bulk RNA-seq data for *ATP6V0E1*, *ATP6V1F*, *ATP6AP2*, and *ATP6V1G1* expression levels using RT-qPCR, we observed marked and consistent downregulation of *ATP6AP2* in therapy-induced senescent MDA-MB-231 (Supplementary Fig. 7d) and MCF-7 (Supplementary Fig. 7e) cells, with a reduction of > 50%. Although all genes showed a trend of declining expression levels in the therapy-treated cells, only *ATP6AP2* expression remained stably downregulated in Doxo-induced senescent cells at 72 and 96 h (Fig. 5f). Treatment with either 500 nM or 250 nM Abe significantly suppressed *ATP6AP2* expression at 72 and 96 h in MDA-MB-231 cells, whereas *ATP6AP2* expression was significantly reduced in MCF-7 cells at 48, 72, and 96 h after exposure to 250 and 500 nM Abe (Fig. 5g and Supplementary Fig. 7f). *ATP6AP2*–a hub gene–is correlated with breast cancer immune infiltration and lysosomal function. Moreover, remarkable suppression of *ATP6AP2* was observed in treatment-induced senescent cells, indicating its potential involvement in modulating cellular biology in the senescence process.

**ATP6AP2 knockdown facilitates intracellular acidification and lysosomal alkalinization in breast cancer cells.** To gain comprehensive insights into the regulatory role of *ATP6AP2* in cellular pH homeostasis and its impact on the senescence process in

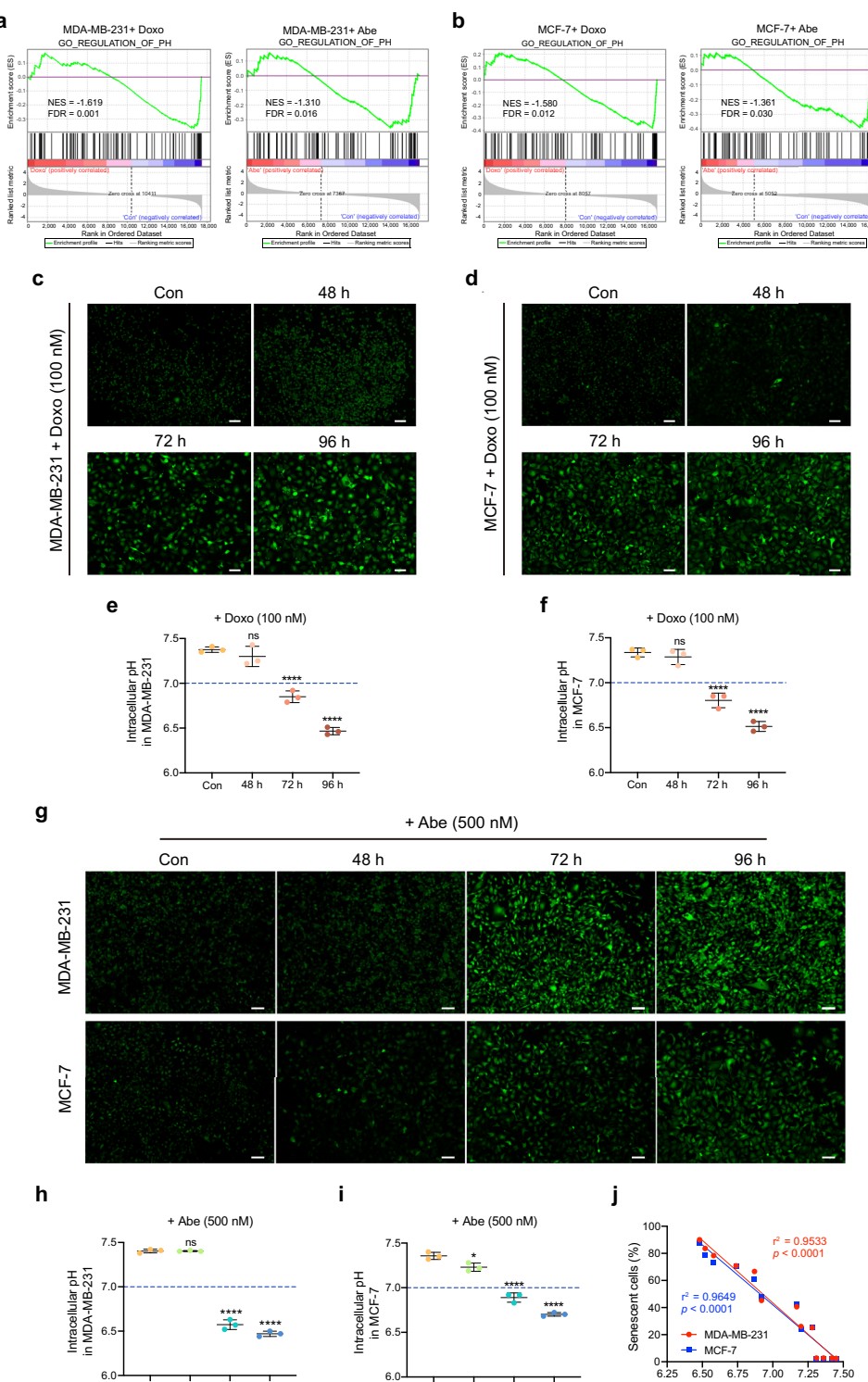

**Fig. 3 Doxo and Abe trigger substantial intracellular acidification in senescent cells. a, b** GSEA analysis was performed with pH-regulated gene set in MDA-MB-231 (**a**) and MCF-7 (**b**) cells, respectively. FDR < 0.05 as the significance threshold. **c, d** Representative fluorescent images of Doxo (100 nM)-treated MDA-MB-231 (**c**) and MCF-7 (**d**) cells stained with pHrodo Green AM. **e, f** Quantitative analysis of the pH$_i$ in each cell line. **g** Representative fluorescent images of Abe (500 nM)-treated MDA-MB-231 (upper panel) and MCF-7 (lower panel) cells stained with pHrodo Green AM. **h, i** Quantitative analysis of the intracellular pH in each cell line. **j** Scatterplot illustrating the correlation between the percentage of senescent cells and pH$_i$ value in breast cancer cells. Scale bars represent 50 μm. Data are shown as the means ± SD of three independent experiments. Data were analyzed by one-way ANOVA with Dunnett's multiple comparisons test (**e, f, h, i**) and simple linear regression with Pearson's correlation analysis (**j**). ns not significant; *$P < 0.05$, ****$P < 0.0001$.

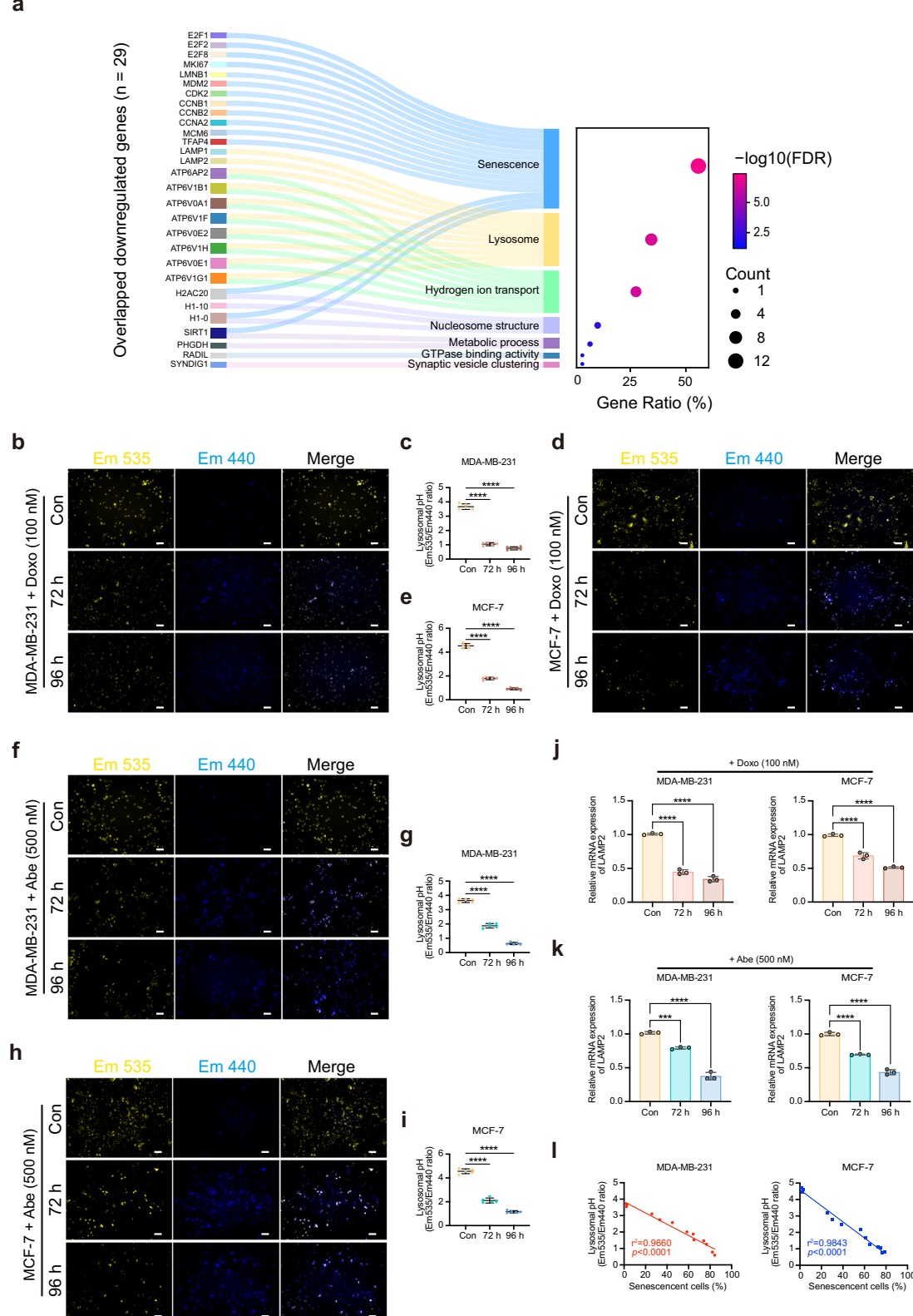

breast cancer cells, we conducted an experimental design employing *ATP6AP2* knockdown depicted in a schematic diagram (Fig. 6a). We first performed RT-qPCR to validate the efficiency of the two distinct siRNA sequences designed to target *ATP6AP2* in comparison with the negative control sequence (si-NC). The results demonstrated a substantial suppression of *ATP6AP2* expression in knockdown breast cancer cells (Fig. 6b).

Next, we performed pH$_i$ detection using pHrodo Green AM staining. The knockdown of *ATP6AP2* exhibited a substantial reduction in pH$_i$, leading to intracellular acidification (Fig. 6c, d). We also determined that the knockdown of *ATP6AP2* resulted in a significant increase in the pH$_L$ status using the LysoSensor Yellow/Blue DND-160 probe (Fig. 6e–h). Notably, the knockdown of *ATP6AP2* triggered a decrease in the number of

**Fig. 4 Doxo and Abe elicit an alkalinized pH$_L$ to cause lysosomal impairment in senescent breast cancer cells. a** Sankey plot showcasing functional enrichment analysis of overlapped downregulated genes ($n = 29$) in the highlighted cluster. The dot plot showed the gene ratio and the total number of genes in each enriched function (FDR < 0.05). **b–e** Representative fluorescent images of Doxo (100 nM)-treated MDA-MB-231 (**b**) and MCF-7 (**d**) cells stained with LysoSensor Yellow/Blue-DND-160 to assess pH$_L$ status. Em 535 nm (yellow); Em 440 nm (blue). The pH$_L$ status was determined ratiometrically in MDA-MB-231 ($n = 5$) (**c**) and MCF-7 ($n = 5$) (**e**) cells. **f–i** Representative fluorescent images of Abe (500 nM)-treated MDA-MB-231 (**f**) and MCF-7 (**h**) cells stained with LysoSensor Yellow/Blue-DND-160 to assess pH$_L$ status. The pH$_L$ status was determined ratiometrically in MDA-MB-231 ($n = 5$) (**g**) and MCF-7 ($n = 5$) (**i**) cells. **j, k** RT-qPCR to determine the relative mRNA expression levels of *LAMP2* in each cell line treated with Doxo (100 nM) (**j**) and Abe (500 nM) (**k**). **l** Scatterplot illustrating the correlation between pH$_L$ and the percentage of senescent cells in breast cancer cells. Scale bars represent 50 μm. Data are shown as the means ± SD of three independent experiments. Fisher's exact test with Benjamini–Hochberg multiple-testing correction (**a**), one-way ANOVA with Dunnett's multiple comparisons test (**c, e, g, i, j, k**), and simple linear regression with Pearson correlation analysis (**l**) was performed. ***$P < 0.001$, ****$P < 0.0001$.

lysosomes, as evidenced by a reduction in visible fluorescent puncta (Fig. 6e, g). Furthermore, we observed a significant decrease in *LAMP2* expression in *ATP6AP2*-knockdown MDA-MB-231 (Fig. 6i) and MCF-7 (Fig. 6j) cells, which suggests that the knockdown of *ATP6AP2* impairs lysosomal function.

To further explore the association between *ATP6AP2* and cellular senescence, we validated the senescence-related phenotypes in breast cancer cells. CCK-8 assay revealed that the knockdown of *ATP6AP2* significantly inhibited the proliferation of both MDA-MB-231 (Supplementary Fig. 8a) and MCF-7 (Supplementary Fig. 8b) cells at 48 and 72 h. Subsequently, SA-β-Gal staining demonstrated a slight increase in the proportion of senescent cells after *ATP6AP2* knockdown (Supplementary Fig. 8c–e). RT-qPCR revealed alterations in the expression of interleukin-1 beta (*IL1B*), chemokine (C-C motif) ligand 2 (*CCL2*), transforming growth factor beta 1 (*TGFB1*), C-X-C motif chemokine ligand 2 (*CXCL2*), and *IL6* in ATP6AP2-knockdown MDA-MB-231 (Supplementary Fig. 8f) and MCF-7 (Supplementary Fig. 8g) cells. Consequently, *ATP6AP2* knockdown triggered a reduction in pH$_i$ and an increase in pH$_L$, resulting in impaired lysosomal function. Moreover, the knockdown of *ATP6AP2* induced cellular senescence in breast cancer cells, accompanied by changes in SASP expression.

**Doxo and Abe induce SASP reprogramming to alter inflammatory and immune profiles in breast cancer cells.** Cellular senescence has been shown to modify the immune status of the tumor microenvironment through the secretion of multiple inflammatory cytokines/chemokines and immune modulators[47,48]. Therefore, we further explored the impact of therapy-induced senescence on the immune and inflammatory processes in breast cancer cells based on the upregulated DEGs identified in the RNA-seq analysis. Venn diagrams highlighted the intersection of 65 overlapping upregulated genes among the four groups of MDA-MB-231 and MCF-7 cells treated with Doxo and Abe (Fig. 7a and Supplementary Data 1–4).

GO enrichment analysis showed that these 65 genes belonged to multiple biological process categories associated with the inflammatory response pathway, cytokine/chemokine-mediated pathways, JAK-STAT cascade, and positive regulation of IL-6/IL-8/IL-10/tumor necrosis factor (TNF) production, which are known pivotal mechanisms of immune regulation (Fig. 7b). Kyoto Encyclopedia of Genes and Genomes (KEGG) pathway analysis further suggested that the overlapped upregulated genes were enriched in several critical immune-related signaling pathways, including cytokine–cytokine receptor interactions, TNF signaling, NF-κB signaling, chemokine signaling, and programmed death-ligand 1 (PD-L1) checkpoint pathways (Fig. 7c).

To investigate the key molecules involved in immune and inflammatory profile changes during therapy-induced senescence, we constructed a PPI network to determine the commonly upregulated genes during therapy-induced senescence in the two breast cancer cell lines clustered into two distinct functional sets —senescence-related genes and immunomodulatory-related genes (Fig. 7d). These results revealed that therapy-induced cellular senescence is involved in immunomodulatory processes. Furthermore, the heatmap demonstrated consistently increased expression of immunomodulation-related genes in the therapy-induced MDA-MB-231 and MCF-7 cells (Fig. 7e and Supplementary Table 3). We selected genes encoding cytokines with relatively high node degrees in the PPI network for validation analysis using RT-qPCR. Indeed, the mRNA expression levels of *CXCL2, TGFB1, CCL2*, and *IL1B* were markedly increased in the breast cancer cell lines after treatment with Doxo and Abe (Supplementary Fig. 9a, b).

Moreover, GSEA indicated a significant correlation between gene sets corresponding to immune-related phenotypes—inflammatory response signaling, interferon-α/interferon-γ response signaling, TNF-α signaling via NF-κB, and the IL6-JAK-STAT3 signaling pathway was enriched in therapy-induced senescent breast cancer cells, respectively (Fig. 7f and Supplementary Fig. 9c). Interestingly, the epithelial–mesenchymal transition (EMT) pathway was significantly enriched in breast cancer cells exposed to Doxo and Abe. Thus, therapy-induced senescence in breast cancer cells triggers the expression of pro-inflammatory molecules via SASP reprogramming, which evokes a profound alteration in the transcriptional profile of genes related to inflammatory and immune regulation.

## Discussion

Dysregulation of the pH status has been observed in senescent human fibroblast cells due to metabolic reprogramming[49]. However, the mechanism underlying the disruption of pH$_i$ homeostasis caused by therapy-induced senescence in breast cancer is not fully understood. Here, we found that Doxo and Abe upregulated the expression of *CDKN1A* and *CDKN2A*, resulting in cellular senescence in breast cancer cells, which is consistent with previous reports[50–53]. Moreover, we demonstrated that these therapy-induced changes in senescent cells were accompanied by a significant downregulation of *ATP6AP2* expression, contributing to disturbances in pH homeostasis that impaired lysosome function and potentially altered immune profiles. We also discovered that senescence-driven SASP reprogramming altered inflammatory and immune transcriptional profiles. Together, these findings suggest that *ATP6AP2* plays a pivotal role in cellular pH regulation during therapy-induced senescence, and that altered pH homeostasis appears to be closely associated with immune profile changes in senescent cancer cells.

Doxo is a widely used chemotherapeutic agent that exerts anticancer effects and induces cellular senescence in different cell types and mouse models[54–59]. Abe causes cell cycle arrest and reduces cell proliferation by inhibiting CDK4/6, thereby contributing to senescence[60–62]. Stable cellular senescence was

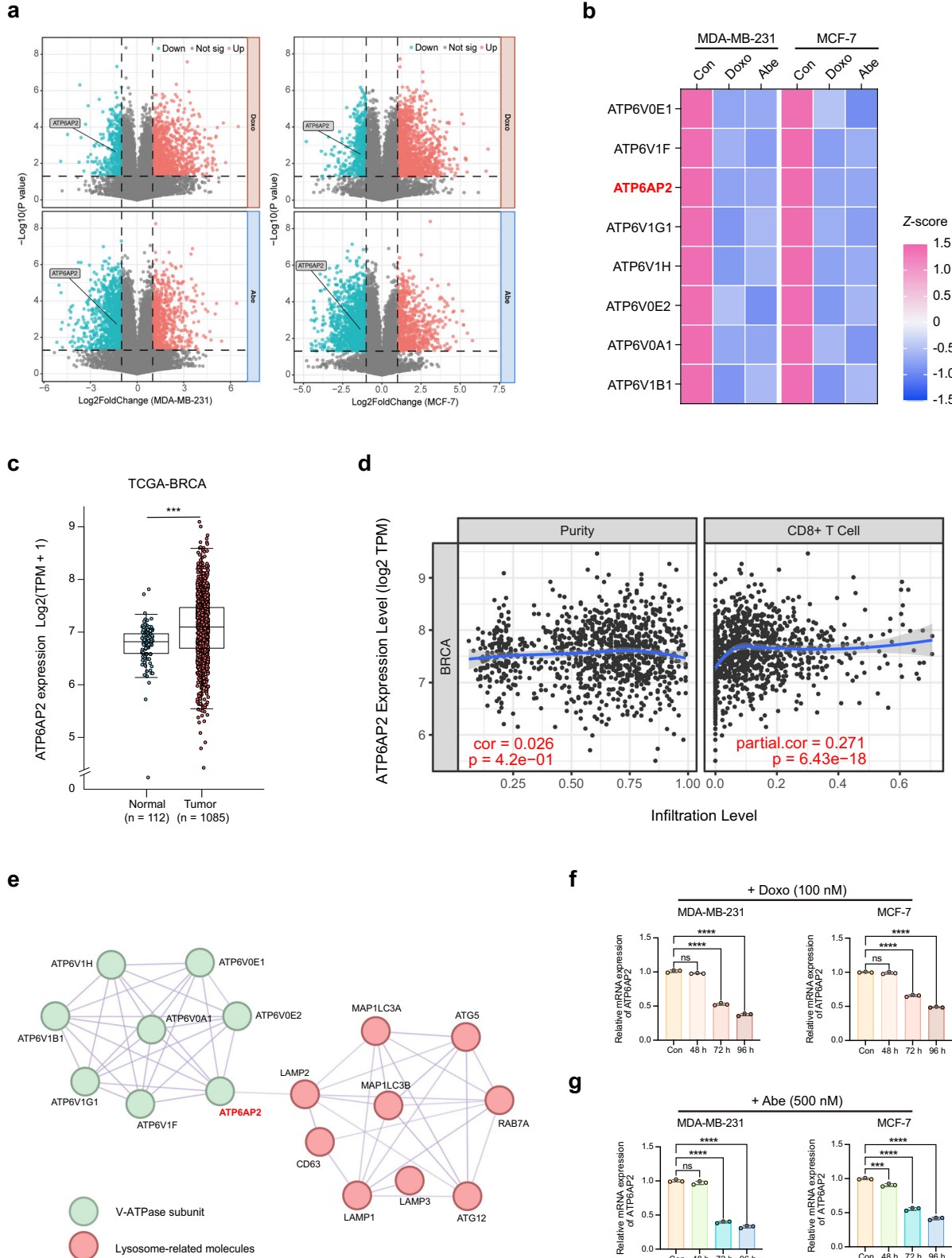

induced in MDA-MB-231 and MCF-7 cells using Doxo and Abe. Furthermore, we found that therapy-induced senescence caused intracellular acidification and lysosomal alkalinization. Based on the bulk RNA-seq analysis of senescent cells compared to non-senescent (untreated) breast cancer cells, we found that the downregulated genes in senescent cells were enriched in V-ATPase activity, affecting $pH_i$ reduction and lysosomal

function. We verified intracellular acidification in senescent cells using pHrodo Green AM staining and lysosomal alkalinization using LysoTracker staining. These findings support the idea that therapy-induced senescence impairs $pH_i$ homeostasis and lyso-somal function[63].

V-ATPase is a protein complex that regulates pH in cells and maintains pH homeostasis within cells and intracellular

**Fig. 5 Doxo and Abe attenuate ATP6AP2 expression in senescent breast cancer cells. a** Volcano plot summarizing differentially expressed genes (DEGs) in therapy-challenged cells vs. control cells. Two vertical lines delineate log2 (fold change) of –1 and 1 boundaries, and the horizontal line displays the statistical significance threshold ($P < 0.05$). The blue and red dots indicate down- and upregulated genes, respectively. **b** Heatmap representing the standardized mRNA abundance values ($z$-scores) of the V-ATPase subunits among overlapped downregulated genes. **c** Relative expression levels of *ATP6AP2* in The Cancer Genome Atlas Breast Invasive Carcinoma (TCGA-BRCA) data. **d** Correlation of *ATP6AP2* with CD8[+] T cell infiltration level in TCGA-BRCA dataset analyzed using Tumor Immune Estimation Resource (TIMER) database. **e** *ATP6AP2* as a hub gene was identified by the protein–protein interaction (PPI) network of differentially expressed V-ATPase subunits and lysosomal-related genes. **f, g** RT-qPCR was used to detect the relative mRNA expression levels of *ATP6AP2* in Doxo (100 nM)-challenged (**f**) and Abe (500 nM)-challenged (**g**) breast cancer cells, respectively. Data are shown as the means ± SD of three independent experiments. Unpaired two-tailed Student's *t* test (**c**) and one-way ANOVA with Dunnett's multiple comparisons test (**f, g**) were performed. ns not significant; ***$P < 0.001$, ****$P < 0.0001$.

organelles, which is essential for lysosomal function[64,65]. The dysregulation of V-ATPase function can lead to pH abnormalities and lysosomal dysfunction[66,67]. We demonstrated that V-ATPase activity was impaired in senescent cells (cells with permanent cell cycle arrest) induced by Doxo and Abe, and that the *ATP6AP2* (V-ATPase subunit) gene was consistently downregulated in these cells. As Baf A1 suppresses lysosomal and autophagic lysosomal functions by blocking V-ATPase activity[68,69], we confirmed the effect of treatment with Baf A1, a compound that disrupts V-ATPase activity. Baf A1 treatment led to more substantial changes in pH in senescent breast cancer cells than treatment with Doxo and Abe. Moreover, Baf A1 treatment resulted in more pronounced intracellular acidification and enhanced lysosomal alkalinization, leading to a decrease in lysosomal abundance, in contrast to the effects observed during therapy-induced senescence.

Through siRNA-mediated knockdown of the *ATP6AP2* gene, we observed a significant increase in intracellular acidification and enhanced lysosomal alkalinization. These changes were accompanied by slight cellular senescence. Similarly, both *ATP6AP2* knockdown and Baf A1 treatment triggered a more substantial pH$_i$ decrease and pH$_L$ increase compared to therapy-induced senescence. Additionally, these phenotypes were accompanied by a severe reduction in lysosomal abundance. Based on our findings, we propose that the substantial impact of *ATP6AP2* knockdown on V-ATPase function is similar to that observed following Baf A1 treatment. However, therapy-induced senescence results in a reduction in *ATP6AP2* expression without complete inhibition of V-ATPase function. Together, these results reveal that *ATP6AP2* suppression plays an essential role in the alteration of cellular pH in senescent cells, potentially due to a lack of proton pumping, which leads to the accumulation of protons in the cytosol, with consequent changes in pH$_i$ and pH$_L$. However, the specific mechanism by which *ATP6AP2* expression is downregulated in therapy-induced senescent cells requires further investigation.

Furthermore, our study demonstrated that the activation of multiple genes associated with the SASP occurs in therapy-induced senescent breast cancer cells. GSEA demonstrated that inflammatory and immune-related pathways were significantly enriched in therapy-induced senescent cells. These findings are consistent with previous studies suggesting that senescent cells actively secrete SASP-related molecules that drive inflammatory responses and modulate immune status[70,71]. Furthermore, GSEA revealed that Doxo- and Abe-induced senescence was enriched in the EMT pathways. These results further substantiate the interplay between cellular senescence and EMT, collectively influencing tumor cell processes[72–74]. Understanding the crosstalk between these pathways can provide valuable insights into the development of more effective and targeted cancer therapies[75]. Interestingly, we found an upregulation of genes in the PD-L1 pathway in therapy-induced senescent cells, which aligns with previous studies indicating PD-L1 accumulation in senescent cells[76,77]. This may be a potential mechanism underlying tumor recurrence and immune evasion following treatment with Doxo

and Abe. Additionally, we discovered an enrichment of *STAT3* expression and related pathways, supporting the relationship between senescence bypass and the immune regulation phenotype[78–80]. However, further investigations are required to fully understand the mechanisms by which senescent cells affect immune regulation.

In conclusion, our findings demonstrated that attenuated *ATP6AP2* expression triggered intracellular acidification (pH$_i$ decrease) and lysosomal alkalinization (pH$_L$ increase) in senescent breast cancer cells exposed to Doxo and Abe. Subsequently, this elevated pH$_L$ contributes to impaired lysosomal function and reduced lysosomal abundance. Additionally, SASP reprogramming in senescent cells upregulates multiple molecules that alter inflammatory and immune gene expression profiles. (Fig. 7g) However, the mechanism by which *ATP6AP2* is attenuated in senescent cells remains unknown. Further research is required to understand the detailed mechanisms of pH dysregulation and SASP reprogramming in senescent cells, and to identify the manner in which dysregulation of intracellular pH homeostasis induced by senescence leads to immunoregulatory gene profile alterations. These findings shed light on the underlying mechanisms of *ATP6AP2* in the interplay among cellular pH regulation, lysosomal function, and immune profiles in the context of Doxo- and Abe-induced senescence in breast cancer cells. Elucidating these underlying mechanisms may have potential implications for the development of targeted therapies and interventions for breast cancer treatment.

## Methods

**Cell lines and cell culture**. MDA-MB-231 (a human triple-negative breast cancer cell line) and MCF-7 (a human luminal A subtype breast cancer cell line) cells were obtained from the American Type Culture Collection (Manassas, VA, US). Cells were maintained in Dulbecco's modified Eagle's medium (DMEM; Nacalai Tesque, Kyoto, Japan) containing 10% fetal bovine serum (FBS; Gibco, Life Technologies, Carlsbad, CA, USA) and 0.01% penicillin/streptomycin (Wako, Osaka, Japan). Both cell lines were cultured at 37 °C in a 5% v/v $CO_2$ atmosphere and used for experiments at early passages (<10 passages). Both cell lines were negative for *Mycoplasma* contamination.

**In vitro senescence induction**. Doxo (Sigma-Aldrich, Darmstadt, Germany) was dissolved in distilled water, and Abe (Verzenio, Eli Lilly, USA) was dissolved in ethanol before use. MDA-MB-231 and MCF-7 cells were cultured under serum-deprived conditions (2% FBS) for 12 h. The cells were then treated with 100 nM Doxo (the concentration determined to not induce severe cell death) and with Abe at a concentration gradient (125, 250, 500, and 1000 nM) for the initial 24 h (1st dose). The culture medium was then replaced with a normal medium (without Doxo or Abe) for 48 h. Subsequently, the cells were re-exposed to the two drugs separately (2nd dose) at the above concentrations and collected at four time points for subsequent experiments.

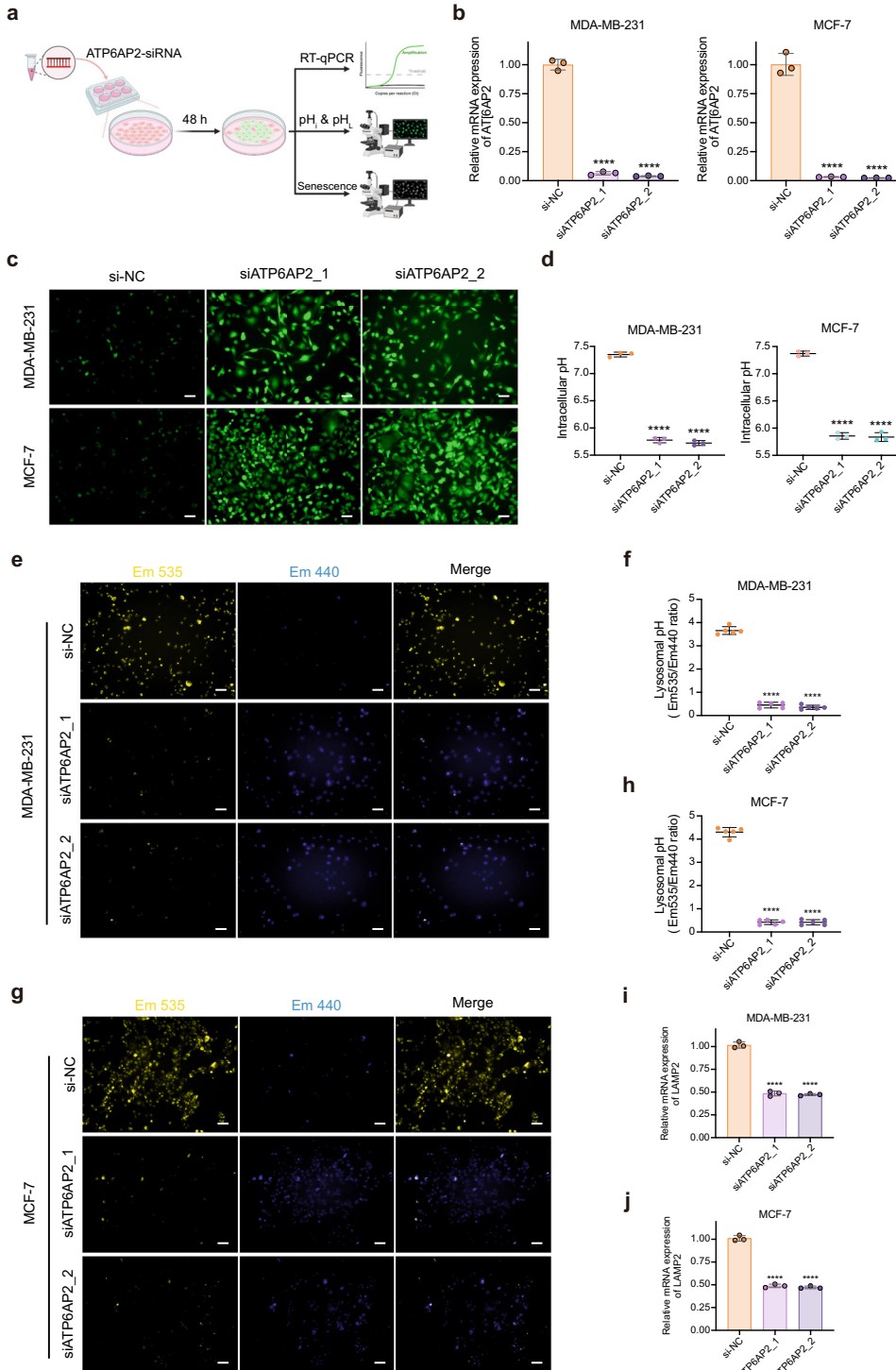

**Fig. 6 ATP6AP2 knockdown facilitates intracellular acidification and lysosomal alkalinization in breast cancer cells. a** Experimental workflow of *ATP6AP2* knockdown (created with BioRender.com). **b** Knockdown efficiency of *ATP6AP2* using siRNA was quantitated by RT-qPCR. **c, d** Representative fluorescent images of breast cancer cells transfected with siATP6AP2 for 48 h ($n = 3$) stained with pHrodo Green AM (**c**) and quantitative analysis of the $pH_i$ (**d**). **e, f** Representative fluorescent images in siATP6AP2 transfected MDA-MB-231 cells ($n = 5$) (**e**) for 48 h stained with LysoSensor Yellow/Blue-DND-160 and quantitative analysis of $pH_L$ status (**f**). **g, h** Representative fluorescent images of $pH_L$ in siATP6AP2 transfected MCF-7 cells ($n = 5$) (**g**) for 48 h stained with LysoSensor Yellow/Blue-DND-160 and quantitative analysis of $pH_L$ status (**h**). **i, j** RT-qPCR analysis of relative mRNA expression levels of *LAMP2* in MDA-MB-231 cells (**i**) and MCF-7 cells (**j**) transfected with siATP6AP2 for 48 h, respectively. Scale bars represent 50 μm. Data are shown as the means ± SD of three independent experiments. One-way ANOVA with Dunnett's multiple comparisons test (**b, d, i, j**) and Brown–Forsythe ANOVA with Dunnett T3 multiple comparisons test (**f, h**) were performed. ****$P < 0.0001$.

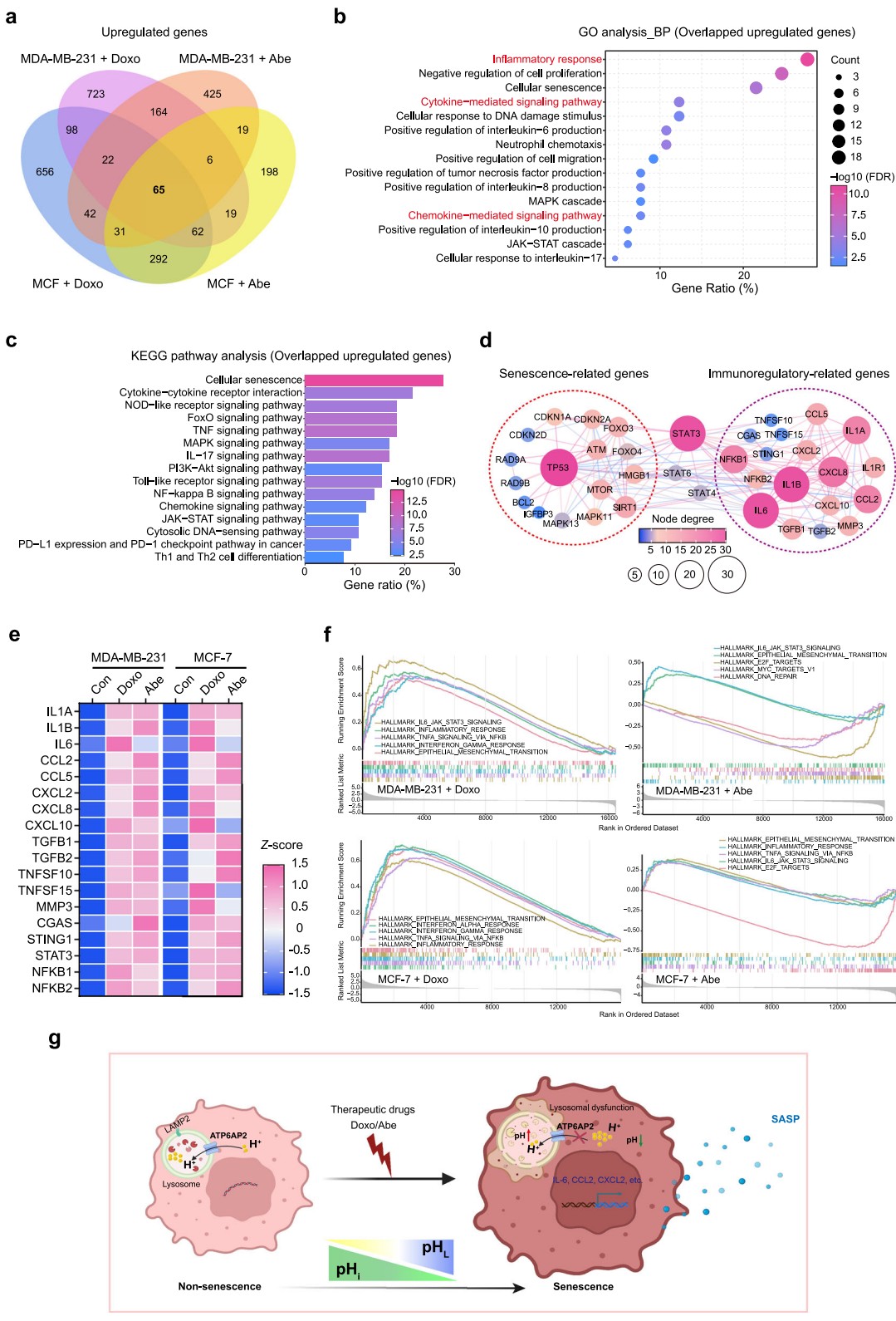

**Cell proliferation assay**. Cell proliferation was evaluated using a CCK-8 assay kit (Dojindo, Kumamoto, Japan) according to the manufacturer's protocol. Cells were seeded in 96-well plates at 3000 cells/well and allowed to attach overnight. The cells were then treated with Doxo or Abe as described above. After collecting the cells at different time points (from 24 to 96 h), 200 μl of medium containing CCK-8 reagent was added to each well,

and the cells were further incubated at 37 °C for 3 h. Absorbance was measured using a SpectraMax 340PC 384 Microplate Reader (Molecular Devices, Tokyo, Japan) at a wavelength of 450 nm.

**Cell cycle assay**. Cell cycle assays were performed using PI/RNase Staining Buffer Kit (BD Biosciences, San Jose, CA, USA). A total

**Fig. 7 Doxo and Abe induce SASP reprogramming to alter inflammatory and immune profiles in breast cancer cells. a** Four-ellipse Venn diagram outlining overlapped upregulated genes in various therapy-treated cells. **b** BP term in GO enrichment analysis of overlapped upregulated genes ($n = 65$). All represented BP terms with FDR < 0.05 are shown. **c** KEGG pathway enrichment analysis of overlapped upregulated genes ($n = 65$). All represented pathway terms with FDR < 0.05 are shown. The color represents the statistical significance of the term. The circle size indicates the counts of enriched genes. **d** STRING network representation of the protein–protein interactions in senescence-related genes (red circles) and immunoregulatory-related genes (purple circles) belonging to overlapped upregulated genes (FDR < 0.05, interaction score ≥ 0.7). The degree of nodes is represented by circle size and color. **e** Heatmap representing the standardized mRNA abundance values ($z$-scores) of the immunoregulatory-related genes. **f** GSEA was performed in the therapy-challenged breast cancer cells using hallmark gene sets. Genes (vertical black lines) represented in gene sets are on the $x$-axis, and the $y$-axis represents the enrichment score (ES). The colored line connects genes and ES points. The colored band shows the degree of correlation of genes with the enriched phenotype FDR < 0.05 as the significance threshold. Fisher's exact test with Benjamini–Hochberg multiple-testing correction (**b**, **c**) was performed. **g** The schematic diagram illustrates the mechanism of pH$_i$ and pH$_L$ regulation by *ATP6AP2* and the immune profile via SASP reprogramming in therapy-induced senescent breast cancer cells (created with BioRender.com).

---

of $3 \times 10^5$ cells were seeded into each well of a 6-well plate and allowed to attach overnight. The cells were harvested after treatment with Doxo or Abe, followed by washing in phosphate-buffered saline and fixation using 70% (v/v) ice-cold ethanol for 1 h (or at 4 °C overnight). The cells were incubated with propidium iodide staining solution for 15 min at room temperature before analysis. Cell cycle distribution was detected immediately by flow cytometry using a BD LSRFortessa Cell Analyzer (BD Biosciences, San Jose, CA, USA), and data were collected using BD FACSDiva Software (v8.0.1, BD Biosciences) and further analyzed using ModFit LT software (v5.0, Verify Software House, Topsham, ME, USA). FACS gating strategy is shown in Supplementary Fig. 10.

**SA-β-Gal assay**. Cellular senescence was determined by the Senescence β-Galactosidase Staining Kit (Cell Signaling Technology, Boston, MA, USA) according to the manufacturer's instructions. A total of $3 \times 10^5$ cells were seeded into each well of a 6-well plate and treated with Doxo or Abe, as described above. The cells were then harvested, fixed with a fixative solution, and incubated with the β-Gal staining solution in a dry incubator without $CO_2$ at 37 °C overnight. After incubation, cells were observed under a BZ-X800 fluorescence microscope (KEYENCE, Osaka, Japan). Quantification of β-Gal-positive senescent cells was performed using Image J software (v1.52, National Institutes of Health, Bethesda, MD, USA).

**RT-qPCR**. Cells were treated with TRIzol reagent (Invitrogen, Carlsbad, CA, USA) and total RNA was extracted using an RNeasy Mini Kit (QIAGEN Sciences, Germantown, MD, USA) following the manufacturer's instructions. cDNA was synthesized by a reverse transcriptase reaction with 500 ng of total RNA using the Transcriptor First Strand cDNA Synthesis Kit (Roche, Basel, Switzerland) and used as a template for qPCR with LightCycler 480 SYBR Green I Master (Roche, Basel, Switzerland) on a StepOnePlus Real-Time PCR System (Applied Biosystems, Foster City, CA, USA). Gene expression levels were normalized to the level of the endogenous control gene *ACTB* using an adjusted $2^{-\Delta\Delta Ct}$ method. The primer sequences used for RT-qPCR are listed in Supplementary Table 4.

**Bulk RNA-seq and gene expression profiling**. Total RNA was isolated using the RNeasy Mini Kit (QIAGEN, Hilden, Germany), following the manufacturer's instructions. The RNA quality was assessed on a Nanodrop DS-11 spectrophotometer (DeNovix, Wilmington, NC, USA) to ensure a 260:280 nm ratio ≥ 2.0, and an RNA integrity number ≥ 9 was assessed by TapeStation RNA Screen Tape (Agilent). RNA-Seq analysis was performed using a transcriptome for targeted next-generation sequencing (Macrogen, Tokyo, Japan). Total RNA (1 µg) was enriched with

polyA + RNA, and sequencing libraries were sequenced with the TruSeq stranded mRNA Library on a NovaSeq6000 platform (Illumina, San Diego, CA, USA). RNA-seq data were checked for quality using the FastQC software (v0.11.9). Sequences were aligned to the human reference genome (GRch38/hg38) using HISAT2(v2.2.1)[81] and sequence reads were assigned to reference genomic features using FeatureCounts[82]. Computational analysis of the RNA-seq data was performed using Galaxy (v22.05.1, The Galaxy platform). Gene counts were scaled and normalized to transcripts per kilobase of millions (TPM) units. The TPM values of each gene were used to calculate the fold-change (FC) and corresponding $p$-values. Differential expression analysis (three biological replicates) was performed using the DESeq2 R package (v1.40.2). Significant DEGs were identified according to the criteria of log2 FC ≥ 1.0 or log2 FC ≤ –1.0 and $P$ value < 0.05. A volcano plot was constructed using the ggplot2 R package (v3.4.2). PCA was performed using the FactoMineR R package (v2.8). The Venn diagrams were constructed using the Venn-Diagram R package (v1.7.3).

**Functional enrichment analysis**. DEGs were subjected to GO and KEGG pathway enrichment analyses using the Database for Annotation, Visualization and Integrated Discovery (DAVID) functional annotation tool[83]. Enriched GO terms and pathways were selected based on the threshold of a false discovery rate (FDR) < 0.05. KEGG pathway and GO were visualized through the clusterProfiler R package (v4.8.2). Heatmaps were generated using the ComplexHeatmap R package (v2.16.0). A Sankey diagram was constructed using the ggsankey R package (davidsjoberg/ggsankey). The PPI networks were constructed using STRING (v11.5, string-db.org/). The edges represent both functional and physical associations among proteins. The results of the PPI network were analyzed and visualized using Cytoscape software (v3.9.1).

**GSEA**. GSEA was conducted using the GSEA (v4.3.2) desktop tool[84]. Hallmark data were acquired from the Molecular Signatures Database (MSigDB, v2022.1)[85]. Gene expression levels of therapy-treated senescent cells versus control cells were used to generate a ranked list file. A total of 1000 permutations were applied to determine the significance of gene set enrichment. The clusterProfiler R package (v4.8.2) was used for GSEA. The enriched phenotypes were considered significant with a $P$ value < 0.01 and FDR < 0.25, and a metric for ranking the genes was obtained by the "Signal to noise" method.

**Baf A1 treatment**. Baf A1 (Bioviotica, Dransfeld, Germany) was dissolved in dimethyl sulfoxide to obtain a 20 µM stock solution. For Baf A1 treatment, cells were cultured in a complete medium containing 200 nM Baf A1 for 3 h before cellular pH detection.

**siRNA knockdown.** For siRNA transfection, siRNAs consisting of Silencer Select Negative Control #1 siRNA (si-NC, UAACGA CGCGACGACGUAAtt), Silencer Select Pre-Designed siRNA s19790 (siATP6AP2_1, GGUCUGUUGUUUUCCGAAAtt), and Silencer Select Pre-Designed siRNA s19792 (siATP6AP2_2, GA GUGUAUAUGGUAGGGAAtt) were purchased from Thermo Fisher Scientific. siRNA was transfected into cells using Lipo-fectamine RNAiMAX reagent (Invitrogen, 13778150) according to the manufacturer's instructions. The cells were seeded in culture plates at 70–80% confluency and subsequently transfected with siRNA for 48 h prior to conducting further experimental analyses.

**Measurement of $pH_i$.** $pH_i$ was determined using the pHrodo Green AM Intracellular pH indicator (Thermo Fisher Scientific, Waltham, MA, USA). Cells grown in DMEM (supplemented with 10% FBS) were seeded in 96-well black plates (5000 cells/well). After Doxo and Abe treatment, the cells were washed with Live Cell Imaging Solution (LCIS; Thermo Fisher Scientific) and labeled with pHrodo Green AM dye for 30 min at 37 °C. After washing with LCIS, cell fluorescence was detected using a BZ-X800 fluorescence microscope (KEYENCE, Osaka, Japan) and analyzed using a SpectraMax GEMINI EM spectrofluorometer (Molecular Devices, Tokyo, Japan) at excitation and emission wavelengths of 509 and 533 nm, respectively. A $pH_i$ standard curve was constructed using intracellular pH calibration buffer Kit (Thermo Fisher Scientific). The cells were treated with different pH calibration solutions (pH 4.5, 5.5, 6.5, and 7.5), and the fluorescence intensity was measured to fit a linear trend line to obtain the pH standard curve.

**$pH_L$ detection.** The $pH_L$ was detected using the ratiometric lysosomal pH dye LysoSensor Yellow/Blue DND-160 (Thermo Fisher Scientific). The cells were seeded in 96-well black plates at a density of 5000 cells/well. After treatment with Doxo and Abe, cells were washed with LCIS and incubated with 2 μM LysoSensor Yellow/ Blue for 5 min at 37 °C under 5% $CO_2$. After washing with LCIS, cell fluorescence was detected and measured using a KEYENCE BZ-X800 fluorescence microscope. The ratio of light emitted at 340 nm and 380 nm excitation was used to determine the $pH_L$ status.

**Statistics and reproducibility.** All results are presented as the mean ± standard deviation (SD) of three independent experiments. Prior to the analysis, a normality test was performed on all collected data. A two-tailed unpaired Student's $t$ test was used for pairwise comparisons, whereas multiple comparisons were evaluated using a one- or two-way analysis of variance, followed by Dunnett's, Sidak's, or Tukey's multiple comparison test. A correlation analysis of two independent samples was performed using Pearson's correlation coefficients. Statistical $P$ values < 0.05 were considered statistically significant. Data analyses and visualizations were conducted using GraphPad Prism software (v9.4.1, GraphPad, La Jolla, CA, USA) and RStudio software (v2023.06.0 + 421, RStudio, MA, USA) in the R computing environment (v4.3.1).

**Reporting summary.** Further information on research design is available in the Nature Portfolio Reporting Summary linked to this article.

## Data availability

The RNA-Seq data generated in this study were deposited in the Gene Expression Omnibus (GEO) database under the accession code GSE222984. Supplementary Data 1–4 comprise the list of DEGs derived from RNA-seq data corresponding to the figures presented in this paper. Source data for generating graphs and charts in the main figures are reported in Supplementary Data 5. Any other data supporting the findings of this study can be made available upon request from the corresponding authors.

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

## Acknowledgements

We thank the Medical Research Support Center (MRSC), Graduate School of Medicine, Kyoto University for providing the research instruments. We thank KEYENCE for providing the fluorescence microscope used in this study. We also thank Masahiro Kawashima, Fengling Pu, Yukiko Fukui, and Yuki Nakamura for their helpful suggestions. This study was supported in part by the Japan Science and Technology Agency Support for Pioneering Research Initiated by the Next Generation (JST-SPRING; grant number JPMJSP2110). This work was also supported by a scholarship from the China Scholarship Council under Grant No.202208050044.

## Author contributions

W.L., K.K., M.T., and E.S. contributed substantially to project design. W.L. performed the experiments and analyzed the data. W.L., C.H., and Y.M. analyzed the RNA-Seq data and interpreted the results. K.K. and S.T. provided suggestions and technical support for the experiments. W.L. drafted the manuscript. K.K. and M.T. supervised the study and revised the manuscript. All authors have approved the submitted version of the manuscript and agreed to be accountable for any part of this work.

## Competing interests

The authors declare the following competing interests: K.K.: grants from TERUMO, Astellas, Eli Lilly, and Kyoto Breast Cancer Research Network; consulting fees from Becton Dickinson, Japan; honoraria from Eisai, Chugai, and Takeda. M.T.: grants from Chugai, Takeda, Pfizer, Kyowa-Kirin, Taiho, JBCRG Associates, Eisai, Eli Lilly, Daiichi-Sankyo, AstraZeneca, Astellas, Shimadzu, Yakult, Nippon Kayaku, AFI Technology, Luxonus, Shionogi, and GL Science; honoraria from Chugai, Takeda, Pfizer, Kyowa-Kirin, Taiho, Eisai, Daiichi-Sankyo, AstraZeneca, Eli Lilly, MSD, Exact Science, Novartis, Konica Minolta, Shimadzu, Yakult, and Nippon Kayaku; advisory board of Kyowa-Kirin, Daiichi-Sankyo, Eli Lilly, Konica Minolta, BMS, Athenex Oncology, Bertis, Terumo, Kansai Medical Net; board of directors of JBCRG Associates, KBCRN, OOTR; Associate Editor of the *British Journal of Cancer*, *Scientific Reports*, *Breast Cancer Research and Treatment*, *Cancer Science*, *Frontiers in Women's Cancer*, *Asian Journal of Surgery*, *Asian Journal of Breast Surgery*; deputy editor of *International Journal of Oncology*. The other authors declare no competing interests.
