## [Peer Review File · Communications Biology]

Reviewers' comments:

Reviewer #1 (Remarks to the Author):

Overall evaluation:

ATP6AP2 has been implicated in human disease. A knockout mice model has shown severe multiple organ function loss in both male and female mice (PMID: 28851918). This manuscript describes that decrease in ATP6AP2 levels are associated with onset of cellular senescence in breast cancer cells upon exposure to chemotherapeutic drugs. The study makes a well-informed association between the levels of ATP6AP2 and intracellular acidification of breast cancer cells exposed to chemotherapy in vitro.

Overall impression of the manuscript:

It appears that authors pin on ATP6AP2 without sufficient premise. It is important to find the mechanism of therapy induced senescence in cancer cells. It seems a gene signature collectively was more indicative of therapy induced senescence rather than pin on single gene. The single gene theory can be pushed while using overexpression data (rescue) of senescence in cancer cells but I don't think it's needed to pin on one gene.

Major Strengths:

Two breast cancer cell lines were used.

Major Weaknesses:

Figure 1: this extensive figure shows the well-known effects of chemo drugs on breast cancer cells. No new information is present in this figure. The information confirms what has been observed in last couple of decades in hundreds of published manuscripts.

Figure 2: The comments to this figure is same as Figure 1.

Figure 3: The RNAseq data described here showed multiple genes including ATP6AP2, ATP6V0A1, ATP6V0E1, ATP6V1B1, ATP6V0E2, ATP6V1H, ATP6V1F, ATP6V1G1 were downregulated upon chemo treatment. Supplement figure 3d,3e shows four genes were significantly downregulated. Others were not shown. Figure 3f shows data for only ATP6AP2 downregulated (others were not tested) and text says only ATP6AP2 was downregulated. How do they know that others were not downregulated? The data for others in Figure 3f needs to be shown. In nutshell, the rationale to pick ATP6AP2 is not articulated well.

I would like to see if over expression of ATP6AP2 would protect the therapy induced senescence and lysosomal dysregulation? If only this gene is considered important.

Reviewer #2 (Remarks to the Author):

The manuscript " Senescence triggers intracellular acidification and lysosomal 1 alkalization via ATP6AP2 attenuation in breast cancer cells", explores the role of ATP6AP2 in doxorubicin and abemaciclib induced senescence in two human cell lines. This is a relevant study because senescence

is well known to play a role in tumor progression, metastasis, and response to therapy. Therefore, investigating the mechanisms that mediate cellular senescence upon chemotherapy treatment is of interest in cancer research. The statistical analysis is appropriate, the materials and methods are well described and the study is interesting, but some critical aspects should be addressed to be considered for publication.

- One major concern that this reviewer has is that the manuscript's novelty appears to be based on the role of ATP6AP2 in mediating the senescent phenotype induced by doxorubicin in MDA-MB-231 and MCF-7 cells (it is known that doxorubicin induces senescence in these cells). While the authors suggest in the abstract that "ATP6AP2-mediated pH regulation during therapy-induced senescence may be linked to immune changes in senescent cancer cells," it would be valuable to perform additional experiments to further support this claim. Specifically, the reviewer recommends that the authors conduct experiments to silence or inhibit ATP6AP2 and determine pHi or pHL and senescence markers, including SASP components. Such experiments could provide more convincing evidence for the proposed mechanism of action and its potential relevance to immune changes in senescent cancer cells.

- Figure 6F and G: please add the color reference or legend for the treatments.

- Please revise the use of "the drugs" (abstract) or "therapeutic drugs (main text) because sometimes it can generate confusion, especially, when only 2 chemotherapeutic compounds were tested.

- Please also reconsider the use of the term "immune response" throughout the manuscript to refer to SASP components, i.e., expression of cytokines, because the immune response was not evaluated in vivo.

Reviewer #3 (Remarks to the Author):

The authors report that ATP6AP2 in senescent breast cancer cells induced by doxorubicin and abemaciclib treatment. ATP6AP2 expression was significantly downregulated in senescent cells, which leads to aberrant pH levels. Moreover, the authors also found that senescent cells showed altered inflammatory and immune transcriptional profiles. These are interesting findings that can help to understand the response of tumours to anti-cancer drugs. However, the study appears to be quite limited both in terms of cells and treatment studied and conclusions.

- My main concern is that the study is limited to MDA-MB-231 and MCF-7 breast cancer cells. It would have been much more interesting to expand this work to other breast cancer types, treatments or study the effect of doxorubicin or abemaciclib in non-cancer cells. Moreover, the authors need to explain why these MDA-MB-231 and MCF-7 and no other cells were used.

- There are some reports that demonstrate the use of doxorubicin (and other cancer treatments) induce senescence in vitro and in vivo in several cancer types including breast cancer, however these studies are not cited. Besides, it would be interesting to know how the findings of the authors might provide insights to these works aiming to eliminate tumours.

- It would also be interesting to know the opinion of the authors about how their findings can help the design of new routes to eliminate senescent cells.

Point-by-point responses to the reviewers' comments

We extend our sincere appreciation to the three reviewers for their invaluable comments and constructive suggestions, which have greatly contributed to enhancing the quality of our study. We are genuinely grateful for their interest in our results. The major revision was to supplement the experimental design by including knockdown experiments, conducting a comprehensive reanalysis of the RNA-seq data, and meticulously restructuring the logical flow of the manuscript to enhance its coherence and clarity. The manuscript has been carefully revised, and point-by-point responses are provided below. We hope that the comments have been addressed in detail. The revised manuscript is marked in red, and the responses are presented in blue.

Structure of the Results section of the manuscript

Original manuscript structure	Revised manuscript structure
1. Doxo and Abe suppress proliferation through cell cycle arrest in breast cancer cells	1. Doxo and Abe promote cellular senescence accompanied by an altered profile of senescence-related genes in breast cancer cells.
2. Doxo and Abe promote cellular senescence accompanied by elevated p16 and p21 expression in breast cancer cells	2. Doxo and Abe induce V-ATPase functional suppression in senescent breast cancer cells
3. Doxo and Abe attenuate ATP6AP2 expression in senescent cells	3. Doxo and Abe trigger substantial intracellular acidification in senescent cells
4. Doxo and Abe trigger substantial intracellular acidification in senescent cells	4. Doxo and Abe elicit an alkalinized pHL to cause lysosomal impairment in senescent breast cancer cells
5. Senescence-induced ATP6AP2 attenuation elicits an alkalinized pHL to cause lysosomal impairment	5. Doxo and Abe attenuate ATP6AP2 expression in senescent breast cancer cells
6. Doxo and Abe induce SASP reprogramming to alter inflammatory and immune profiles in breast cancer cells	6. ATP6AP2 knockdown facilitates intracellular acidification and lysosomal alkalinization in breast cancer cells
	7. Doxo and Abe induce SASP reprogramming to alter inflammatory and immune profiles in breast cancer cells

Reviewers' comments:

Reviewer #1 (Remarks to the Author):

Overall evaluation:

ATP6AP2 has been implicated in human disease. A knockout mice model has shown severe multiple organ function loss in both male and female mice (PMID: 28851918). This manuscript describes that decrease in ATP6AP2 levels are associated with onset of cellular senescence in breast cancer cells upon exposure to chemotherapeutic drugs. The study makes a well-informed association between the levels of ATP6AP2 and intracellular acidification of breast cancer cells exposed to chemotherapy in vitro.

Response: Thank you for your careful review and insightful suggestions regarding this manuscript. We have included the references you kindly provided within the Introduction section (*Lines 62–64 in the revised manuscript: ATP6AP2 is a vital protein involved in fundamental cellular processes, and its ablation results in impaired viability due to multiple organ deficiencies*³⁴).

References:

34 Wendling, O., et al. *Atp6ap2* ablation in adult mice impairs viability through multiple organ deficiencies. *Sci Rep.* 7, 9618 (2017).

Overall impression of the manuscript:

It appears that authors pin on ATP6AP2 without sufficient premise. It is important to find the mechanism of therapy induced senescence in cancer cells. It seems a gene signature collectively was more indicative of therapy induced senescence rather than pin on single gene. The single gene theory can be pushed while using overexpression data (rescue) of senescence in cancer cells but I don't think it's needed to pin on one gene.

Response: We strongly agree with your point of view, which correctly emphasizes the significant downregulation of V-ATPase subunits during cellular senescence, leading to alterations in cellular pH homeostasis. Indeed, our research demonstrates that inducing cellular senescence through treatment results in the downregulation of specific V-ATPase subunits. Furthermore, utilization of 200 nM bafilomycin A1 (Baf A1), a V-ATPase-specific inhibitor, has shown notable effects on intracellular pH (pH_i) and lysosomal pH (pH_L) elevation. We have made corresponding revisions to the manuscript (*Lines 146–173 in the revised manuscript, see revised Fig. 2*).

Some studies have shown that ATP6AP2, an accessory subunit, is essential for the biogenesis of active V-ATPase. Moreover, ATP6AP2 is primarily localized on the membrane of organelles and contributes to the maintenance of pH homeostasis within intracellular compartments. (*Lines 53–59 in the revised manuscript*). To further explore the regulatory role of ATP6AP2 as a hub gene in pH homeostasis in doxorubicin (Doxo)- and abemaciclib (Abe)-induced senescent cells, we conducted RNA-seq analysis and performed ATP6AP2 gene knockdown experiments. We aim to provide robust support for our hypotheses and reinforce the strength of our findings in response to the reviewers' comments.

revised Fig. 2b, d, f and h

Lines 146–173 in the revised manuscript:

Doxo and Abe induce V-ATPase functional suppression in senescent breast cancer cells.

To explore the phenotypic changes induced by Doxo and Abe treatment in senescent breast cancer cells, we compared the transcriptional alterations between senescent and control cells. Among the DEGs, 29 were found to be commonly downregulated in both cell lines, as shown in the Venn diagram in Fig. 2a. Subsequently, we performed Gene Ontology (GO) enrichment analysis to determine the biological functions of the downregulated genes. GO enrichment analysis revealed that these genes were predominantly enriched in the GO biological processes (BP) “cellular senescence” and “intracellular pH reduction,” in the cellular compartment (CC) “vacuolar proton-transporting V-type ATPase complex,” and in the molecular function (MF) “proton-transporting ATPase activity, rotational mechanism” (Fig. 2b). Therefore, we hypothesized that therapy-induced cellular senescence causes a decrease in V-ATPase, which consequently contributes to pHi decline.

To ascertain whether V-ATPase attenuates the therapy-triggered pHi decrease in breast cancer cells, we used 200 nM bafilomycin A1 (Baf A1), a specific V-ATPase inhibitor, to treat MDA-MB-231 and MCF-7 cells over a time gradient. After cells were subjected to 1 h of 200 nM Baf A1 treatment, the pHi was observed to be lower than that of control cells, and a significant difference in pHi was detected after 3 h of inhibitor treatment (Supplementary Fig. 4a). Notably, 200 nM Baf A1 treatment did not affect cell viability (Supplementary Fig. 4b). Fluorescence microscopy confirmed that the pHi decreased under the same treatment conditions (Fig. 2c and d).

As shown in Fig 2b, the overlapping downregulated genes were also enriched in the GO BP term associated with lysosomal membrane. To further elucidate whether V-ATPase attenuated the Doxo- and Abe-induced increase in pHL, we assayed pHL in breast cancer cells treated with 200 nM Baf A1 for 3 h. Baf A1 significantly decreased yellow fluorescence (emission [Em] 535 nm) intensity, with a corresponding increase in blue fluorescence (Em 440 nm) intensity in both MDA-MB-231 (Fig. 2e and f) and MCF-7 (Fig. 2g and h) cells. This shift in the fluorescence spectra resulted in a substantial decrease in the ratio of yellow to blue fluorescence intensity, indicating an increase in pHL. These consistent changes in the fluorescence profiles were observed in both MDA-MB-231 and MCF-7 cells following Baf A1 treatment. Therefore, Doxo and Abe impede the function of V-ATPase, thereby inducing the dysregulation of cellular pH homeostasis in senescent breast cancer cells.

Lines 53–59 in the revised manuscript

Adenosine triphosphatase H⁺ transporting accessory protein 2 (ATP6AP2), also known as the prorenin receptor, is an essential accessory subunit for the biogenesis of active vacuolar-type adenosine triphosphatase (V-ATPase)¹⁸⁻²¹. V-ATPases are complex multi-subunit enzymes that function as rotary nanomotors that pump protons and maintain intracellular pH (pHi) homeostasis²²⁻²⁴. V-ATPase defects perturb autophagy in various systems²⁵. ATP6AP2 is predominantly localized in the lysosome and plasma membrane, and plays a crucial role in energy conservation and acidification of intracellular compartments to support cellular biological activity^{26, 27}.

Major Strengths:

Two breast cancer cell lines were used.

Response: We thank the reviewer for this comment, which has affirmed our work.

Major Weaknesses:

Figure 1: this extensive figure shows the well-known effects of chemo drugs on breast cancer cells. No new information is

present in this figure. The information confirms what has been observed in last couple of decades in hundreds of published manuscripts.

Response: We totally agree with your comments. We acknowledge that there have been numerous studies on chemotherapeutic drug-induced cellular senescence. However, since we modified the methods of inducing senescence using Doxo and Abe, we aimed to select more appropriate concentrations and durations for the induction of a stable senescent phenotype through therapeutic drugs. We conducted a series of selective experiments to determine the concentrations and durations for Doxo and Abe treatment. We have placed all the results related to validating cell proliferation suppression and cell cycle arrest in the supplementary materials (*see revised Supplementary Fig. 1 and 2*) and have made the corresponding modifications to the manuscript (*Lines 99–110 in the revised manuscript*).

Lines 99–110 in the revised manuscript:

The Cell Cycle Kit-8 (CCK-8) assay showed that Doxo significantly suppressed the proliferation of both MDA-MB-231 and MCF-7 cells (Supplementary Fig. 1a). Abe inhibited cell proliferation in time- and dose-dependent manners (Supplementary Fig. 1b). However, 1000 nM Abe caused massive cell death, and 24 h of treatment was insufficient to suppress proliferation; thus, this concentration of Abe was deemed unsuitable for senescence induction in breast cancer cells. To investigate whether proliferation was suppressed through cell cycle arrest under our treatment conditions, we performed flow cytometry to analyze cell cycle distribution in breast cancer cells. Doxo induced G2 cell cycle arrest in MDA-MB-231 cells, in contrast to the G1 phase arrest observed in MCF-7 cells (Supplementary Fig. 1c and d). Conversely, Abe induced G1 cell cycle arrest in both MDA-MB-231 and MCF-7 cells at various concentrations (Supplementary Fig. 2a-f). These results suggested that Doxo and Abe substantially inhibited breast cancer cell proliferation, an essential feature of cellular senescence, through cell cycle arrest in a time- and dose-dependent manner.

Figure 2: The comments to this figure is same as Figure 1.

Response: We completely agree with your comment. In accordance with your suggestion, we have exclusively included the senescence-associated β -galactosidase (SA- β -Gal) staining and RT-qPCR results from breast cancer cells treated with Doxo and Abe (at a concentration of 500 nM) in the revised Figure 1b-g. As for the additional Abe-treated groups at different concentrations, we have placed the relevant information in the Supplementary Materials (*Lines 112–133 in the revised manuscript, see revised Supplementary Fig. 3*).

To ensure a rigorous analysis and comprehensive description of the senescence induced by Doxo and Abe, we conducted a thorough reanalysis of the RNA-seq data, further confirming the generation of the senescent phenotype and profiling of genes associated with senescence (*Lines 134–144 in the revised manuscript, see revised Fig. 1h-k*). This comprehensive approach addresses the reviewers' concerns and strengthens our findings.

revised Fig. 1h-k

Lines 112–133 in the revised manuscript:

Doxo triggered significant cellular senescence after 48 h of exposure and showed more pronounced staining with further exposure, particularly after 96 h of treatment (Fig. 1b and d). Similarly, treatment with 500 nM Abe increased the proportion of senescent cells and caused significant cellular senescence after 96 h of exposure (Fig. 2c and e). Both 250 and 125 nM Abe induced cellular senescence, but the effect was more pronounced at 250 nM, where the proportion of senescent cells was higher than that in the 125 nM treatment group, which showed a relatively lower proportion of senescent cells (less than 50%). (Supplementary Fig. 3a-d). Moreover, the microscopic images demonstrated that the cell morphology became irregular, and that cell size was enlarged in the senescent cells compared to that in the control non-senescent cells.

Considering the essential roles of numerous molecular expression profiles correlated with the senescence phenotype, we next sought to verify the molecules considered hallmarks of cellular senescence. Reverse transcription-quantitative polymerase chain reaction (RT-qPCR) revealed that Doxo simultaneously increased the mRNA levels of cyclin-dependent kinase inhibitor 1A (CDKN1A), cyclin-dependent kinase inhibitor 2A (CDKN2A), matrix metalloproteinase-3 (MMP3), and interleukin-6 (IL6), accompanied by decreased laminin subunit beta-1 (LMNB1) levels, with a significant increase detected after 96 h of treatment (Fig. 2f). Similarly, 500 nM Abe induced significant changes in the expression of senescence-related molecules, with the most prominent effect observed after 96 h of exposure (Fig. 2g). Moreover, 250 nM Abe treatment yielded molecular changes with a similar trend to 500 nM Abe treatment (Supplementary Fig. 3f), whereas alterations in the expression levels of senescence-associated molecules induced by 125 nM Abe treatment were unstable (Supplementary Fig. 3e). Thus, 125 nM Abe treatment was excluded from subsequent experiments.

Lines 134–144 in the revised manuscript:

We further analyzed and profiled the gene expression of therapy-induced cellular senescence using bioinformatic methods applied to bulk RNA-seq. Principal component analysis (PCA) revealed an obvious difference in a distinct cluster of differentially expressed senescence-related genes between therapy-challenged and untreated breast cancer cells (Fig. 1h). Additionally, Gene Set Enrichment Analysis (GSEA) indicated that the Doxo and Abe treatment groups were enriched in the “FRIDMAN_SENESCENCE-UP” gene set (Fig. 1i and j). The heatmap illustrates the expression patterns of senescence-related genes (using the “REACTOME_CELLULAR_SENESCENCE” gene set) obtained from bulk RNA-seq analysis (Fig. 1k). Overall, these results demonstrated that Doxo and Abe trigger stable cellular senescence by simultaneously upregulating CDKN1A and CDKN2A expression. Furthermore, the senescent phenotype became increasingly evident with prolonged

treatment, particularly after 96 h.

Figure 3: The RNAseq data described here showed multiple genes including ATP6AP2, ATP6V0A1, ATP6V0E1, ATP6V1B1, ATP6V0E2, ATP6V1H, ATP6V1F, ATP6V1G1 were downregulated upon chemo treatment. Supplement figure 3d,3e shows four genes were significantly downregulated. Others were not shown. Figure 3f shows data for only ATP6AP2 downregulated (others were not tested) and text says only ATP6AP2 was downregulated. How do they know that others were not downregulated? The data for others in Figure 3f needs to be shown. In nutshell, the rationale to pick ATP6AP2 is not articulated well.

Response: Thank you for your insightful comments. We selected V-ATPase subunits (ATP6V0E1, ATP6V1F, ATP6AP2, ATP6V1G1, ATP6V1H, ATP6V0E2, ATP6V0A1, and ATP6V1B1) from the 29 downregulated genes. The heatmap demonstrates a consistent downregulation trend in the expression of various subunits (*see revised Fig. 5b*). By analyzing the baseline expression of different V-ATPase subunits using RNA-seq data, we identified relatively higher baseline expression levels of ATP6V0E1, ATP6V1F, ATP6AP2, and ATP6V1G1 (*see revised Supplementary Fig. 7a and b*). Because we are aware that the inhibition of V-ATPase function can disrupt intracellular pH homeostasis, we carefully considered the baseline expression levels of the overlapping downregulated genes. We found that ATP6V1H, ATP6V0E2, ATP6V0A1, and ATP6V1B1 exhibited relatively low basal expression, to the extent that studying their inhibition and impact on V-ATPase may not be sufficient. Therefore, we validated the suppression of mRNA expression of the top four subunits (ATP6V0E1, ATP6V1F, ATP6AP2, and ATP6V1G1) after 96 h of treatment with Doxo and Abe. Only ATP6AP2 exhibited consistent inhibition, exceeding 50% in both treatment groups (*see revised Supplementary Fig. 7d and e*).

Furthermore, we also discovered a positive correlation between ATP6AP2 and CD8⁺ T cell infiltration in breast cancer, which was more significant than that of the other subunits (*see revised Fig. 5d and Supplementary Fig. 7c*). Importantly, protein-protein interaction (PPI) analysis revealed that only ATP6AP2 interacted with LAMP2 (*see revised Fig. 5e*). Therefore, we can consider ATP6AP2 as a hub gene involved in the regulation of pH_i and pH_L in senescent breast cancer cells and may affect the immune cell infiltration. We have made appropriate revisions to the relevant sections of the manuscript in response to your feedback (*Lines 232–266 in the revised manuscript*).

revised Fig. 5b, d and e

revised Supplementary Fig. 7a and b

revised Supplementary Fig. 7c-e

Lines 232–266 in the revised manuscript:

Doxo and Abe attenuate ATP6AP2 expression in senescent breast cancer cells.

We further confirmed that the hub gene was responsible for compromised V-ATPase activity and disrupted cellular pH homeostasis. The volcano plot illustrates the DEGs identified in the bulk of the gene expression matrix. Notably, ATP6AP2 was consistently significantly downregulated in the Doxo- and Abe-treated groups (Fig. 5a). Intriguingly, in both MDA-MB-231 and MCF-7 cells, the majority of DEGs were upregulated following treatment with Doxo, but downregulated after treatment with Abe, compared to the expression levels in control cells. We first analyzed the expression levels of V-ATPase subunits among DEGs in non-senescent cells and found a relatively higher baseline expression of ATP6V0E1, ATP6V1F, ATP6AP2, and ATP6V1G1 (Supplementary Fig. 7a and b). Subsequently, we visualized the expression profile of the differentially expressed V-ATPase subunits using a heatmap (Fig. 5b and Supplementary Table 2).

We identified a significant upregulation in ATP6AP2 expression among the tumor groups by analyzing The Cancer Genome Atlas Breast Invasive Carcinoma (TCGA-BRCA) data (Fig. 5c). Moreover, we observed a positive correlation between ATP6AP2 and CD8⁺ T cell tumor immune infiltration through the analysis of the Tumor Immune Estimation Resource (TIMER) database (Fig. 5d). Analysis of the top four subunits of V-ATPase based on their basal expression levels revealed that ATP6V0E1 and ATP6V1F exhibited relatively weak positive correlations with CD8⁺ T cell infiltration in tumors.

Conversely, ATP6V1F negatively correlated with CD8⁺ T cell infiltration (Supplementary Fig. 7c).

To further elucidate the association between ATP6AP2 and lysosomal function, we performed a protein–protein interaction (PPI) analysis involving differentially expressed V-ATPase subunits and lysosome-related genes. The analysis revealed significant interactions among various differentially expressed V-ATPase subunits. Notably, ATP6AP2 displayed a pronounced association with LAMP2, indicating the potential functional relevance between ATP6AP2 and lysosomal processes (Fig. 5e).

Following the validation of bulk RNA-seq data for ATP6V0E1, ATP6V1F, ATP6AP2, and ATP6V1G1 expression levels using RT-PCR, we observed marked and consistent downregulation of ATP6AP2 in therapy-induced senescent MDA-MB-231 (Supplementary Fig. 7d) and MCF-7 cells (Supplementary Fig. 7e), with a reduction of > 50%. Although all genes showed a trend of declining expression levels in the therapy-treated cells, only ATP6AP2 expression remained stably downregulated in Doxo-induced senescent cells at 72 and 96 h (Fig. 5f). Treatment with either 500 nM or 250 nM Abe significantly suppressed ATP6AP2 expression at 72 and 96 h in MDA-MB-231 cells, whereas ATP6AP2 expression was significantly reduced in MCF-7 cells at 48, 72, and 96 h after exposure to 250 and 500 nM Abe (Fig. 5g and Supplementary Fig. 7f). ATP6AP2—a hub gene—is correlated with breast cancer immune infiltration and lysosomal function. Moreover, remarkable suppression of ATP6AP2 was observed in treatment-induced senescent cells, indicating its potential involvement in modulating cellular biology in the senescence process.

I would like to see if over expression of ATP6AP2 would protect the therapy induced senescence and lysosomal dysregulation? If only this gene is considered important.

Response: Thank you for this comment. We agree that rescue experiments will be useful in understanding the important role of ATP6AP2 in pH regulation in senescent breast cancer cells. Our study examined the induction of significant and stable senescence phenotypes in breast cancer cells using therapeutic drugs (Doxo and Abe). However, it is very difficult to establish a cell line with stable ATP6AP2 overexpression through a lentivirus in permanently proliferation-suppressed cells, which differs from gene-induced senescence, where simultaneous modulation of multiple genes is feasible. To provide a more thorough understanding of the underlying mechanisms of ATP6AP2, we used siRNA to knock down ATP6AP2 expression in non-senescent breast cancer cell lines (non-senescence). ATP6AP2 knockdown resulted in significant intracellular acidification and lysosomal alkalization, which resulted in the suppression of LAMP2 expression. (*see revised Fig. 6*). Additionally, ATP6AP2 knockdown partially enhanced breast cancer cell senescence and modulated the senescence-associated secretory phenotype (SASP; *see revised Supplementary Fig. 8*). We appreciate your consideration of these experimental complexities and the comprehensive efforts we have undertaken to elucidate the role of ATP6AP2 in our study.

Reviewer #2 (Remarks to the Author):

The manuscript "Senescence triggers intracellular acidification and lysosomal pH alkalinization via ATP6AP2 attenuation in breast cancer cells", explores the role of ATP6AP2 in doxorubicin and abemaciclib induced senescence in two human cell lines. This is a relevant study because senescence is well known to play a role in tumor progression, metastasis, and response to therapy. Therefore, investigating the mechanisms that mediate cellular senescence upon chemotherapy treatment is of interest in cancer research. The statistical analysis is appropriate, the materials and methods are well described and the study is interesting, but some critical aspects should be addressed to be considered for publication.

Response: We thank you for your careful review and insightful suggestions regarding this manuscript.

- One major concern that this reviewer has is that the manuscript's novelty appears to be based on the role of ATP6AP2 in mediating the senescent phenotype induced by doxorubicin in MDA-MB-231 and MCF-7 cells (it is known that doxorubicin induces senescence in these cells). While the authors suggest in the abstract that "ATP6AP2-mediated pH regulation during therapy-induced senescence may be linked to immune changes in senescent cancer cells," it would be valuable to perform additional experiments to further support this claim. Specifically, the reviewer recommends that the authors conduct experiments to silence or inhibit ATP6AP2 and determine pH_i or pH_L and senescence markers, including SASP components. Such experiments could provide more convincing evidence for the proposed mechanism of action and its potential relevance to immune changes in senescent cancer cells.

Response:

1. Thank you for your insightful comments. Our study primarily investigates the mechanisms underlying ATP6AP2-triggered intracellular pH (pH_i) reduction and lysosomal pH (pH_L) elevation in doxorubicin (Doxo) and abemaciclib (Abe)-induced senescent breast cancer cells. We observed that senescent cells with disrupted pH homeostasis exhibited a reprogrammed senescence-associated secretory phenotype (SASP), leading to increased expression of inflammatory and immune-related genes. This finding prompted us to speculate that there may be a connection between this phenotype and immune regulation. We agree with the reviewer that it is necessary to provide solid support for this hypothesis. Therefore, we conducted a comprehensive reanalysis of the RNA-seq data, revealing a clear association and interaction between molecules related to aging and those involved in immune regulation (*see revised Fig. 7d*). Senescent cells were significantly enriched in pathways related to inflammation and immune regulation (*see revised Fig. 7f*). In addition, we confirmed the expression of certain inflammatory factors (CXCL2, TGFB1, CCL2, and IL1B) by RT-qPCR (*see revised Supplementary Fig. 9a and b*). We have made corresponding revisions in the Abstract (*Lines 28–30 in the revised manuscript: These findings suggest a potential link between ATP6AP2-mediated cellular pH regulation and immunoregulation during therapy-induced senescence in breast cancer cells.*). Revision to address these results (*Lines 303–328 in the revised manuscript*). In the Discussion section, we have highlighted the need for further research to explore the correlation between cellular pH regulation and alterations in immune-related gene profiles. (*Lines 393–397 in the revised manuscript: However, the mechanism by which ATP6AP2 is attenuated in senescent cells remains unknown. Further research is required to understand the detailed mechanisms of pH dysregulation and SASP reprogramming in senescent cells, and to identify the manner in which dysregulation of intracellular pH homeostasis induced by senescence leads to immunoregulatory gene profile alterations.*).

2. Following the reviewer's concern regarding the need for enhanced evidence to confirm the regulation of pH and immune profiling changes in senescent breast cancer cells, we performed siRNA-mediated knockdown experiments targeting ATP6AP2. We established that the knockdown of ATP6AP2 resulted in intracellular acidification (pH_i decrease; *see revised*

Fig. 6c-d) and lysosomal alkalization (pH_L increase), and triggered lysosomal impairment (see revised Fig. 6e-j). Furthermore, the knockdown of ATP6AP2 lead to the generation of a cellular senescent phenotype and altered the expression of SASP-related molecules (see revised Supplementary Fig. c and e-g). We have incorporated these additional results into the Results section (Lines 268–294 in the revised manuscript).

revised Fig. 7d and f

revised Supplementary Fig. 9a and b

revised Fig. 6c-d

revised Fig. 6e-j

revised Supplementary Fig. c and e-g

Lines 303–328 in the revised manuscript:

GO enrichment analysis showed that these 65 genes belonged to multiple biological process categories associated with the inflammatory response pathway, cytokine/chemokine-mediated pathways, JAK-STAT cascade, and positive regulation of IL-6/IL-8/IL-10/tumor necrosis factor (TNF) production, which are known pivotal mechanisms of immune regulation (Fig. 7b). Kyoto Encyclopedia of Genes and Genomes (KEGG) pathway analysis further suggested that the overlapped upregulated genes were enriched in several critical immune-related signaling pathways, including cytokine–cytokine receptor interactions, TNF signaling, NF-κB signaling, chemokine signaling, and programmed death-ligand 1 (PD-L1) checkpoint pathways (Fig. 7c).

To investigate the key molecules involved in immune and inflammatory profile changes during therapy-induced senescence,

we constructed a PPI network to determine the commonly upregulated genes during therapy-induced senescence in the two breast cancer cell lines clustered into two distinct functional sets: senescence-related genes and immunomodulatory-related genes (Fig. 7d). These results revealed that therapy-induced cellular senescence is involved in immunomodulatory processes. Furthermore, the heatmap demonstrated consistently increased expression of immunomodulation-related genes in the therapy-induced MDA-MB-231 and MCF-7 cells (Fig. 7e and Supplementary Table 3). We selected genes encoding cytokines with relatively high node degrees in the PPI network for validation analysis using RT-qPCR. Indeed, the mRNA expression levels of CXCL2, TGFB1, CCL2, and IL1B were markedly increased in the breast cancer cell lines after treatment with Doxo and Abe (Supplementary Fig. 9a and b).

Moreover, GSEA indicated a significant correlation between gene sets corresponding to immune-related phenotypes — inflammatory response signaling, interferon- α /interferon- γ response signaling, TNF- α signaling via NF- κ B, and the IL6-JAK-STAT3 signaling pathway was enriched in therapy-induced senescent breast cancer cells, respectively (Fig 7f and Supplementary Fig. 9c). Interestingly, the epithelial-mesenchymal transition (EMT) pathway was significantly enriched in breast cancer cells exposed to Doxo and Abe. Thus, therapy-induced senescence in breast cancer cells triggers the expression of pro-inflammatory molecules via SASP reprogramming, which evokes a profound alteration in the transcriptional profile of genes related to inflammatory and immune regulation.

Lines 268–294 in the revised manuscript:

ATP6AP2 knockdown facilitates intracellular acidification and lysosomal alkalization in breast cancer cells.

To gain comprehensive insights into the regulatory role of ATP6AP2 in cellular pH homeostasis and its impact on the senescence process in breast cancer cells, we conducted an experimental design employing ATP6AP2 knockdown depicted in a schematic diagram (Fig. 6a). We first performed RT-qPCR to validate the efficiency of the two distinct siRNA sequences designed to target ATP6AP2 in comparison with the negative control sequence (si-NC). The results demonstrated a substantial suppression of ATP6AP2 expression in knockdown breast cancer cells (Fig. 6b).

Next, we performed pHi detection using pHrodo Green AM staining. The knockdown of ATP6AP2 exhibited a substantial reduction in pHi, leading to intracellular acidification (Fig. 6c and d). We also determined that the knockdown of ATP6AP2 resulted in a significant increase in the pHL status using the LysoSensor Yellow/Blue DND-160 probe (Fig. 6e-h). Notably, the knockdown of ATP6AP2 triggered a decrease in the number of lysosomes, as evidenced by a reduction in visible fluorescent puncta (Fig. 6e and g). Furthermore, we observed a significant decrease in LAMP2 expression in ATP6AP2-knockdown MDA-MB-231 (Fig. 6i) and MCF-7 (Fig. 6j) cells, which suggests that the knockdown of ATP6AP2 impairs lysosomal function.

To further explore the association between ATP6AP2 and cellular senescence, we validated the senescence-related phenotypes in breast cancer cells. CCK-8 assay revealed that the knockdown of ATP6AP2 significantly inhibited the proliferation of both MDA-MB-231 (Supplementary Fig. 8a) and MCF-7 (Supplementary Fig. 8b) cells at 48 and 72 h. Subsequently, SA- β -Gal staining demonstrated a slight increase in the proportion of senescent cells after ATP6AP2 knockdown (Supplementary Fig. 8c-e). RT-qPCR revealed alterations in the expression of interleukin-1 beta (IL1B), chemokine (C-C motif) ligand 2 (CCL2), transforming growth factor beta 1 (TGFB1), C-X-C motif chemokine ligand 2 (CXCL2), and IL6 in ATP6AP2-knockdown MDA-MB-231 (Supplementary Fig. 8f) and MCF-7 (Supplementary Fig. 8g) cells. Consequently, ATP6AP2 knockdown triggered a reduction in pHi and an increase in pHL, resulting in impaired lysosomal function. Moreover, the knockdown of ATP6AP2 induced cellular senescence in breast cancer cells, accompanied by changes in SASP expression.

- Figure 6F and G: please add the color reference or legend for the treatments.

Response: Thank you for this comment. We have made the necessary revisions to the figure as per the reviewer's request.

- Please revise the use of “the drugs” (abstract) or “therapeutic drugs (main text) because sometimes it can generate confusion, especially, when only 2 chemotherapeutic compounds were tested.

Response: Thank you for this comment. As suggested, we corrected all “the drugs” or “therapeutic drugs” to “Doxo and Abe.”

- Please also reconsider the use of the term "immune response" throughout the manuscript to refer to SASP components, i.e., expression of cytokines, because the immune response was not evaluated in vivo.

Response: Thank you for raising this important point. We apologize for the confusion between the concepts of immune response and immune-related gene expression (immune gene profiles). Per the reviewer's suggestion, we have made the necessary revisions to address this issue.

Reviewer #3 (Remarks to the Author):

The authors report that ATP6AP2 in senescent breast cancer cells induced by doxorubicin and abemaciclib treatment. ATP6AP2 expression was significantly downregulated in senescent cells, which leads to aberrant pH levels. Moreover, the authors also found that senescent cells showed altered inflammatory and immune transcriptional profiles. These are interesting findings that can help to understand the response of tumours to anti-cancer drugs. However, the study appears to be quite limited both in terms of cells and treatment studied and conclusions.

Response: We thank you for your careful review and insightful suggestions regarding this manuscript, and for the thorough and constructive comments. We appreciate your consideration of the intriguing aspects of this study. Our study aimed to investigate the molecular mechanisms underlying cellular pH homeostasis regulation and alterations in immune gene profiles in senescent breast cancer cells treated with doxorubicin (Doxo) and abemaciclib (Abe) at the cellular level. We understand the reviewer's perspective and acknowledge the importance of further exploration of direct molecular mechanisms and their relevance in the clinical treatment of breast cancer. As discussed in the Discussion section, we discovered that therapy-induced senescent breast cancer cells were enriched in PD-L1 and EMT-related pathways, which warrants further in-depth investigation (*Lines 378–385 in the revised manuscript*). We aimed to explore the foundational groundwork through current cellular-level basic research, laying the initial groundwork for future investigations at animal and clinical levels.

Lines 378–385 in the revised manuscript:

Furthermore, GSEA revealed that Doxo- and Abe-induced senescence was enriched in the EMT pathways. These results further substantiate the interplay between cellular senescence and EMT, collectively influencing tumor cell processes^{72–74}. Understanding the crosstalk between these pathways can provide valuable insights into the development of more effective and targeted cancer therapies⁷⁵. Interestingly, we found an upregulation of genes in the PD-L1 pathway in therapy-induced senescent cells, which aligns with previous studies indicating PD-L1 accumulation in senescent cells^{76,77}. This may be a potential mechanism underlying tumor recurrence and immune evasion following treatment with Doxo and Abe.

- My main concern is that the study is limited to MDA-MB-231 and MCF-7 breast cancer cells. It would have been much more interesting to expand this work to other breast cancer types, treatments or study the effect of doxorubicin or abemaciclib in non-cancer cells. Moreover, the authors need to explain why these MDA-MB-231 and MCF-7 and no other cells were used.

Response:

1. Thank you for raising these important points. Our primary focus was to investigate the effects of therapeutic drugs (Doxo and Abe) which are commonly used breast cancer treatments on breast cancer cells, specifically exploring phenotypic changes in senescent breast cancer and molecular profiling. It is important to note that we have not yet optimized the effect of these therapeutic drugs on normal cells (non-tumor cells); therefore, we did not observe any senescence-related changes or cellular processes in normal cells. We acknowledge the reviewer's feedback and will consider this aspect for future research.

2. Regarding the reviewer's comment about the selection of the MDA-MB-231 and MCF-7 cell lines for our study, we would like to clarify the rationale behind our selection. Doxorubicin is a frontline chemotherapeutic agent that is frequently used to treat triple-negative breast cancer (TNBC) in both clinical practice and basic studies^{1,2,3,4}. MDA-MB-231 cells, derived from patients with metastatic TNBC⁵, represent a more clinically relevant model that aligns well with the clinical context. Abemaciclib is commonly used for the clinical treatment of ER-positive (ER+) and HER2-negative (HER2-) breast cancer^{6,7,8,9}. It is important to note that MCF-7 breast cancer cells align with this molecular subtype, being ER-positive and HER2-

negative^{10, 11}. This choice of cell line provides a relevant model for studying the effects of abemaciclib, particularly in the context of ER+ and HER2- breast cancer. MDA-MB-231 and MCF-7 cell lines encompass two distinct breast cancer subtypes: TNBC and Luminal. For HER2-positive breast cancer, there are specific targeted therapies (trastuzumab, an anti-HER2 antibody). Therefore, we believe that the selection of the MDA-MB-231 and MCF-7 cell lines for our study of therapy-induced senescence was more appropriate.

- There are some reports that demonstrate the use of doxorubicin (and other cancer treatments) induce senescence in vitro and in vivo in several cancer types including breast cancer, however these studies are not cited. Besides, it would be interesting to know how the findings of the authors might provide insights to these works aiming to eliminate tumours.

Response:

1. Thank you for your insightful suggestions and comments. We apologize for not referring to a relevant study. We have added a discussion in the Discussion section regarding the research on Doxo- and Abe-induced cellular senescence, and have included the relevant references (*Lines 341–343 in the revised manuscript: Doxo is a widely used chemotherapeutic agent that exerts anticancer effects and induces cellular senescence in different cell types and mouse models^{54–59}. Abe causes cell cycle arrest and reduces cell proliferation by inhibiting CDK4/6, thereby contributing to senescence^{60–62}.*).

2. In response to the reviewer's comment, we have provided insights into tumor elimination. We observed the disruption of pH homeostasis and immune-related gene changes in senescent breast cancer cells. Dysregulation of cellular pH can have a significant impact on the overall tumor elimination process (*Lines 67–70 in the revised manuscript: Recent studies have suggested that an impaired pH is a hallmark of cancer³⁸. Dysregulated pH dynamics facilitate various cancer cell behaviors such as cell proliferation, migration, metastasis, evasion of apoptosis, and metabolic adaptation^{39–41}. Additionally, an acidified intracellular environment suppresses antibody-dependent cytotoxicity in breast cancer cells⁴².*). The positive correlation between ATP6AP2 expression and CD8⁺ T cell infiltration in breast cancer suggests a potential role for ATP6AP2 in influencing the immune response within the tumor microenvironment (*see revised Fig. 5d*). Significant downregulation of ATP6AP2 in therapy-induced senescent cells could potentially disrupt tumor elimination. In conclusion, our findings suggest that significant downregulation of ATP6AP2 in therapy-induced senescent cells disrupts cellular pH regulation. Moreover, ATP6AP2 inhibition triggered alterations in immune-related molecular expression (*revised Supplementary Fig. 8f and g*). Indeed, the combined effects we observed offer insights for further investigation of immune-mediated cell elimination mechanisms.

revised Fig. 5d

d

It would also be interesting to know the opinion of the authors about how their findings can help the design of new routes to eliminate senescent cells.

Response: Thank you for your insightful comments. We extend our gratitude to the reviewer for prompting us to consider directions for future research. In this study, we identified ATP6AP2 in senescence-associated pH dysregulation and its impact on immune-related changes in breast cancer cells. Downregulation of ATP6AP2 inhibited cell proliferation and induced a cellular senescence phenotype (*see revised Supplementary Fig. 5a-c*). Altering cellular ATP6AP2 expression could potentially influence the clearance of senescent cells.

ATP6AP2 interacts with various important signaling pathways and molecules. These include MAPK/ERK, PI3K-AKT-mTOR, TGF- β 1, and Wnt/ β -catenin signaling pathway^{12, 13, 14, 15, 16}. These pathways are associated with cellular processes, including cell growth, differentiation, survival, and senescence. The intricate connections between ATP6AP2 and these pathways suggest a potential role for ATP6AP2 in the regulation of cellular senescence, which could affect the clearance of senescent cells.

Furthermore, our study demonstrated that senescent breast cancer cells with pH dysregulation undergo SASP (senescence-associated secretory phenotype) reprogramming, resulting in the induction of the expression of various immune regulatory factors (*see revised Fig. 7e*). Further exploration of whether senescent cells with a dysregulated cellular pH can promote immune cell-mediated clearance within the immune microenvironment through the release of cytokines is also a promising route.

Investigating the paracrine effects of senescent cells and their potential to induce metabolic reprogramming in nearby senescent or normal cells, leading to the removal of senescent cells via metabolic alternation¹⁷, is a highly intriguing direction to pursue. Understanding the interplay between the ATP6AP2-induced dysregulation of intracellular pH and metabolic reprogramming in senescent cells may offer valuable insights into the mechanisms that govern cellular senescence. These insights may uncover novel strategies for the selective elimination of senescent cells, which is a significant goal in senescence-related studies and cancer therapy.

References in Rebuttal Letter

- O'Brien ME, *et al.* Reduced cardiotoxicity and comparable efficacy in a phase III trial of pegylated liposomal doxorubicin HCl (CAELYX/Doxil) versus conventional doxorubicin for first-line treatment of metastatic breast cancer. *Ann Oncol* **15**, 440-449 (2004).
- Masuda N, *et al.* Adjuvant Capecitabine for Breast Cancer after Preoperative Chemotherapy. *N Engl J Med* **376**, 2147-2159 (2017).
- Shafei A, *et al.* A review on the efficacy and toxicity of different doxorubicin nanoparticles for targeted therapy in metastatic breast cancer. *Biomed Pharmacother* **95**, 1209-1218 (2017).
- Monteran L, *et al.* Chemotherapy-induced complement signaling modulates immunosuppression and metastatic relapse in breast cancer. *Nat Commun* **13**, 5797 (2022).
- Cailleau R, Olivé M, Cruciger QV. Long-term human breast carcinoma cell lines of metastatic origin: preliminary characterization. *In Vitro* **14**, 911-915 (1978).
- Goetz MP, *et al.* MONARCH 3: Abemaciclib As Initial Therapy for Advanced Breast Cancer. *J Clin Oncol* **35**, 3638-3646 (2017).
- Sledge GW, Jr., *et al.* The Effect of Abemaciclib Plus Fulvestrant on Overall Survival in Hormone Receptor-Positive, ERBB2-Negative Breast Cancer That Progressed on Endocrine Therapy-MONARCH 2: A Randomized Clinical Trial. *JAMA Oncol* **6**, 116-124 (2020).
- Harbeck N, *et al.* Adjuvant abemaciclib combined with endocrine therapy for high-risk early breast cancer: updated efficacy and Ki-67 analysis from the monarchE study. *Ann Oncol* **32**, 1571-1581 (2021).
- Rugo HS, *et al.* Adjuvant abemaciclib combined with endocrine therapy for high-risk early breast cancer: safety and patient-reported outcomes from the monarchE study. *Ann Oncol* **33**, 616-627 (2022).
- Soule HD, Vazquez J, Long A, Albert S, Brennan M. A human cell line from a pleural effusion derived from a breast carcinoma. *J Natl Cancer Inst* **51**, 1409-1416 (1973).
- Lee AV, Oesterreich S, Davidson NE. MCF-7 cells--changing the course of breast cancer research and care for 45 years. *J Natl Cancer Inst* **107**, (2015).
- Nguyen G, Delarue F, Burcklé C, Bouzhir L, Giller T, Sraer JD. Pivotal role of the renin/prorenin receptor in angiotensin II production and cellular responses to renin. *J Clin Invest* **109**, 1417-1427 (2002).

13. Ichihara A, Yatabe MS. The (pro)renin receptor in health and disease. *Nat Rev Nephrol* **15**, 693-712 (2019).
14. Cruciat CM, *et al.* Requirement of prorenin receptor and vacuolar H⁺-ATPase-mediated acidification for Wnt signaling. *Science* **327**, 459-463 (2010).
15. Hoffmann N, Peters J. Functions of the (pro)renin receptor (Atp6ap2) at molecular and system levels: pathological implications in hypertension, renal and brain development, inflammation, and fibrosis. *Pharmacological Research* **173**, 105922 (2021).
16. Krop M, Lu X, Danser AHJ, Meima ME. The (pro)renin receptor. A decade of research: what have we learned? *Pflügers Archiv - European Journal of Physiology* **465**, 87-97 (2013).
17. Johmura Y, *et al.* Senolysis by glutaminolysis inhibition ameliorates various age-associated disorders. *Science* **371**, 265-270 (2021).

REVIEWERS' COMMENTS:

Reviewer #1 (Remarks to the Author):

The authors have addressed all the critiques in a responsible and clear manner. The manuscript is acceptable in its current format.

Reviewer #2 (Remarks to the Author):

Authors have effectively addressed all my concerns, resulting in a significantly improved article.